# STYLE WALTZ: DANCING BETWEEN CONTENT AND STYLE IN FACE STYLIZATION

## ABSTRACT

Achieving precise artistic control while preserving identity remains a central challenge in facial image stylization, with most methods requiring costly training and offering limited flexibility. This paper introduces **StyleBrush**, a training-free stylization framework grounded in Riemannian geometry, which resolves this tension through a principled, dual-control optimization. Our core theoretical contribution is to reframe style transfer as a geodesic path-finding problem on a latent manifold. By leveraging the pullback metric, we establish a local isometry that validates optimizing a path's energy in the embedding space as a means to approximate true geodesics, providing a rigorous foundation for style interpolation. This geometric framework is uniquely applied at two critical stages of the diffusion process: first, for interpolating content and style latents to ensure a semantically continuous fusion, and second, for modulating query features in self-attention layers to dynamically control stylization intensity. The unification of these two control mechanisms under a single geometric principle constitutes the primary novelty of our approach, enabling fine-grained and theoretically-grounded stylization control without any model training. Empirical validation on standard benchmarks confirms that our method significantly outperforms existing state-of-the-art approaches across a suite of quantitative and qualitative metrics.

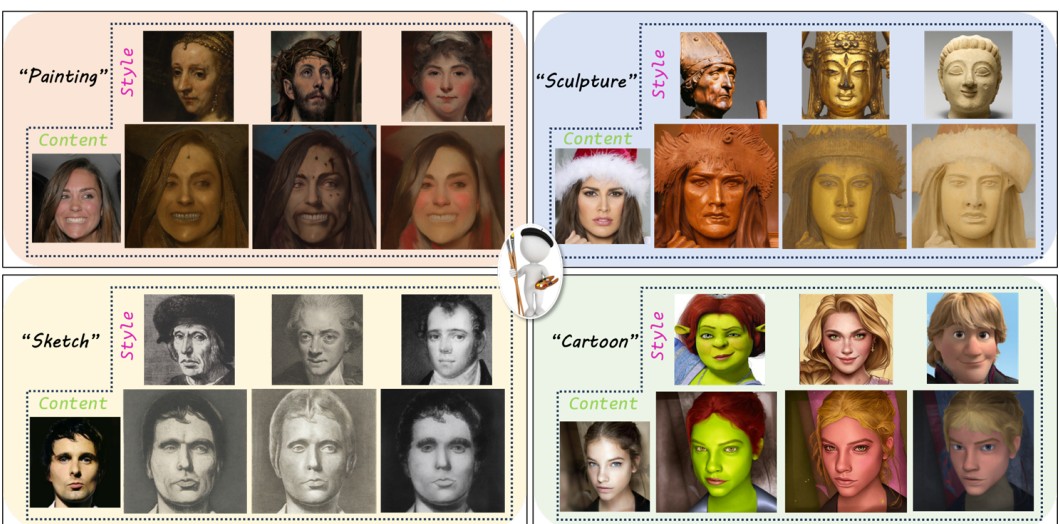

## 1 INTRODUCTION

Image stylization, which aims to transform images into artistic styles while preserving their original content characteristics Gatys et al. (2016); Kwon & Ye (2022a); Deng et al. (2022); Zhang et al. (2023b); Hong et al. (2023); Zhang et al. (2022); An et al. (2021); Yang et al. (2023), has attracted significant research attention in recent years. While traditional style transfer methods have shown success with general images, facial image stylization presents unique challenges that require special consideration. While artistic style transfer is a well-established field, the specific challenge of stylizing facial images while maintaining identity has received surprisingly limited research attention in recent

years Khowaja et al. (2024); Yi et al. (2020); Shi et al. (2019); Chen et al. (2020). The key challenge of facial image stylization lies in striking a delicate balance - the stylization must maintain recognizable facial features that preserve the subject's identity while simultaneously achieving compelling artistic effects (Cohen et al., 2025). This requires more precise control over the stylization process compared to general image stylization, making it a particularly demanding problem in the field.

Recent advances in diffusion models have demonstrated remarkable capabilities in various image generation and manipulation tasks Ho et al. (2020); Nichol & Dhariwal (2021); Kim et al. (2022); Wang et al. (2023a); Kong et al. (2023), with promising applications in style transfer Jeong et al. (2024); Wang et al. (2023b); Chung et al. (2024); Kwon & Ye (2022b). While these models excel at producing high-quality stylized images through their iterative denoising process, several critical limitations persist in existing diffusion-based approaches. First, adapting these models to new artistic styles typically requires extensive computational resources and training time Zhang et al. (2023b). Second, they often lack precise control mechanisms over the stylization process, making it challenging to achieve desired artistic effects Chung et al. (2024). Third, we found these approaches frequently struggle to maintain an optimal equilibrium between faithful style transfer and the preservation of essential identity features, particularly crucial for facial stylization tasks.

To address these challenges, we propose a novel training-free approach **StyleBrush** that offers precise control over the face stylization process. First, we develop a linear style interpolation mechanism that leverages geodesic paths Shao et al. (2018), the shortest paths between two points on a curved surface, to enable careful control over the initial content-style fusion. By computing the geodesic path between the content and style images in the latent space, we can sample different points along this mathematically optimal trajectory, providing fine-grained control over style integration while ensuring smooth and natural transitions. Second, we implement a dynamic style injection control mechanism that adaptively modulates the balance between content and style features during the diffusion process. Building upon attention-based style injection Chung et al. (2024), we compute a geodesic path between the content query and style query in the attention space. By sampling points along this path at different stages of diffusion, we can precisely control how much style information is injected into the content features. This dynamic control allows for stronger style injection in areas that benefit from artistic expression while maintaining more conservative injection rates in regions crucial for identity preservation, such as facial features.

The main contributions of our work can be summarized as follows:

- A novel training-free method for facial stylization that can be generalized across various stable diffusion architectures, achieving state-of-the-art performance while providing fine-grained control over the stylization process
- A dual-control framework combining geodesic-based style interpolation and adaptive style injection, enabling precise content-style fusion through optimal trajectories in latent space while dynamically optimizing the balance between artistic stylization and identity preservation via attention-based geodesic sampling
- Extensive empirical evaluation and rigorous experimental analysis demonstrating statistically significant performance improvements across a comprehensive suite of quantitative metrics and systematic qualitative assessments

## 2 RELATED WORK

### 2.1 IMAGE STYLE TRANSFER

Image style transfer has been an active research area in computer vision. Early approaches like Gatys et al. Gatys et al. (2016) pioneered neural style transfer by using CNN features to separate and recombine content and style. Since then, numerous methods have been proposed to improve style transfer quality and efficiency Johnson et al. (2016); Huang & Belongie (2017). Recent transformer-based methods like StyTr2 Deng et al. (2022) leverage self-attention mechanisms for better feature transformation. CLIPStyler Kwon & Ye (2022a) enables text-guided style transfer by utilizing CLIP's multimodal representations.

With the advent of diffusion models Nichol & Dhariwal (2021), several works have explored their potential for style transfer. Wang et al. Wang et al. (2023b) proposed StyleDiffusion for controllable disentangled style transfer. Zhang et al. Zhang et al. (2023b) developed an inversion-based approach

for more precise style control. Domain-aware methods like Zhang et al. (2022) incorporate contrastive learning to better capture domain-specific style features.

Recent works have also focused on improving artistic quality while maintaining content fidelity. AesPA-Net Hong et al. (2023) introduces aesthetic pattern awareness for more visually pleasing results. ArtFlow An et al. (2021) employs reversible neural flows to achieve unbiased style transfer. Yang et al. Yang et al. (2023) proposed a zero-shot contrastive loss for text-guided diffusion style transfer, demonstrating the potential of diffusion models in this domain.

## 2.2 FACIAL IMAGE STYLIZATION

Recognizing the unique challenges of facial stylization, researchers have developed specialized approaches for face images Khowaja et al. (2024). Early attempts adapted general style transfer methods with face-specific constraints Selim et al. (2016). More recent works have incorporated facial landmark detection Zhang et al. (2020) and identity preservation mechanisms. StyleGAN-based approaches Karras et al. (2019; 2020b) and unsupervised image-to-image translation methods have shown promise in facial stylization tasks like selfie-to-anime conversion. However, these methods typically require task-specific training to achieve satisfactory results. While many existing approaches excel at single-style transfer, they often struggle with multi-style fusion and customized style combinations. Methods like DreamBooth Ruiz et al. (2022) and Textual Inversion Gal et al. (2022) face similar challenges when attempting to combine multiple artistic styles like cartoon, anime, and arcane aesthetics. A key limitation across these approaches is their difficulty in preserving facial identity features, poses, and characteristics during style transfer. The misalignment between source and target domain facial features frequently results in artifacts that compromise the quality of the stylized output.

Our work advances the field by introducing a training-free approach that achieves fine-grained control over both the initial style fusion and the ongoing style injection process in diffusion models. Unlike previous methods that require extensive training or offer limited control, our dual-control framework enables precise stylization while maintaining facial identity, addressing key limitations in existing approaches.

## 3 METHOD

### 3.1 FROM LIMITATIONS TO GEODESICS: INTUITIVE MOTIVATION

Existing style transfer methods struggle with a fundamental dilemma Chung et al. (2024); Deng et al. (2022); Wang et al. (2023b); Xu et al. (2025): achieving effective style transfer while preserving content identity. Training-based approaches can produce high-quality results but sacrifice flexibility—requiring expensive retraining for each new style domain. Training-free methods offer flexibility but often fail at the core task: they either over-stylize (losing facial identity) or under-stylize (weak style transfer), and frequently produce artifacts such as ghosting and blurred features. This limitation persists across both CNN-based and diffusion-based approaches, suggesting a deeper issue beyond model architecture.

Our key insight addresses this fundamental challenge through geometric principles. The limitation stems from treating the latent space as flat Euclidean space where simple interpolation suffices. In reality, diffusion models learn a curved manifold where valid images reside. By navigating along geodesic paths—the natural curves on this manifold—we achieve three critical properties simultaneously: (1) smooth semantic transitions that avoid low-probability regions and artifacts, (2) automatic identity preservation through the manifold's geometric structure without explicit constraints, and (3) controllable style strength via the interpolation parameter. However, since latent space compresses fine-grained details, we complement geodesic navigation with feature-space injection during decoding. This dual control—geodesics for semantic consistency and feature injection for style details—enables training-free style transfer that resolves the flexibility-quality dilemma.

## 3.2 THEORETICAL FOUNDATION

Basically, we want to transfer the problem in coordinate space $\mathcal{X}$ to a Riemannian manifold $\mathcal{M}$ through the mapping $g$. For a general background on geodesic path computation, see Appendix B. Before introducing the geodesic-guided style interpolation in Section 3.3, we have the theoretical results that guarantee the feasibility of our method. First, we will define and analyze the corresponding Riemannian structures and geodesic distances in $\mathcal{X}$ and $\mathcal{M}$.

**Definition 3.1.** Let $\gamma : [0, 1] \to \mathcal{X}$ be an absolutely continuous (AC) curve. The arc length of $\gamma$ with respect to the pullback Riemannian metric $\mathrm{g}$ is

$$L_{(\mathcal{X},\mathrm{g})}(\gamma) \;=\; \int_0^1 \left\| \dot{\gamma}(t) \right\|_{g,\gamma(t)} dt \;=\; \int_0^1 \sqrt{\dot{\gamma}(t)^\top G\big(\gamma(t)\big)\, \dot{\gamma}(t)}\; dt.$$

Let $\eta(t) = g(\gamma(t)) \in \mathcal{M}$. The arc length of $\eta$ with respect to the metric induced by the ambient Euclidean inner product is

$$L_{(\mathcal{M},\|\cdot\|_2)}(\eta) \;=\; \int_0^1 \left\| \dot{\eta}(t) \right\|_2 dt \;=\; \int_0^1 \left\| \tfrac{d}{dt}\big(g \circ \gamma\big)(t) \right\|_2 dt.$$

**Geodesic distances.** For $x_1, x_2 \in \mathcal{X}$, the geodesic distance induced by $\mathrm{g}$ is

$$d_{\mathrm{g}}(x_1, x_2) \;:=\; \inf \Big\{ L_{(\mathcal{X},\mathrm{g})}(\gamma) \;:\; \gamma \in AC([0,1], \mathcal{X}),\; \gamma(0) = x_1,\; \gamma(1) = x_2 \Big\}.$$

For $y_1, y_2 \in \mathcal{M}$, the geodesic distance induced by the Euclidean metric restricted to $\mathcal{M}$ is

$$d_{\mathcal{M}}(y_1, y_2) \;:=\; \inf \Big\{ L_{(\mathcal{M},\|\cdot\|_2)}(\eta) \;:\; \eta \in AC([0,1], \mathcal{M}),\; \eta(0) = y_1,\; \eta(1) = y_2 \Big\}.$$

The following proposition exhibits the local isometry between $(\mathcal{X}, \mathrm{g})$ and $(\mathcal{M}, \|\cdot\|_2)$. The proof can be found in Appendix.

**Proposition 3.2** (Local isometry of $(\mathcal{X}, \mathrm{g})$ into $(\mathcal{M}, \|\cdot\|_2)$)**.** *Let $g : \mathcal{X} \to \mathbb{R}^p$ be continuously differentiable and assume $J_g(x)$ has full column rank at $x \in \mathcal{X}$. Consequently, there exists a neighborhood $U \ni x$ such that $g|_U : U \to \mathcal{M}_U$ is a Riemannian isometry onto its image (lengths of curves are preserved), i.e.,*

$$L_{(\mathcal{X},\mathrm{g})}(\gamma) = L_{(\mathcal{M}_U,\|\cdot\|_2)}(g \circ \gamma) \quad \textit{for all absolutely continuous } \gamma : [0,1] \to U.$$

*Proof.* By definition of the pullback metric, $\mathrm{g}_x(u, v) = \langle J_g(x)u,\, J_g(x)v \rangle_2$, so $J_g(x)$ preserves inner products on tangent spaces. Full rank implies $J_g(x)$ is injective; by the inverse function theorem there is a neighborhood $U$ of $x$ on which $g$ is a $C^1$ diffeomorphism onto its image $\mathcal{M}_U$. For any absolutely continuous $\gamma : [0,1] \to U$,

$$\left\| \dot{\gamma}(t) \right\|_{g,\gamma(t)} = \left\| J_g(\gamma(t))\dot{\gamma}(t) \right\|_2 = \left\| \tfrac{d}{dt}\big(g \circ \gamma\big)(t) \right\|_2,$$

hence lengths are preserved after integration. This is precisely the definition of a local Riemannian isometry onto $\mathcal{M}_U$ endowed with the Euclidean induced metric. □

The local isometry proposition ensures *geometric fidelity*: optimizing arc length, tracing geodesics, or taking Riemannian gradients in latent space $\mathcal{X}$ with the pullback metric is identical to doing so on the generated manifold $\mathcal{M}$. Practically, it justifies the straight-line length $\|g(x_{k+1}) - g(x_k)\|_2$ between two points in the embedding space $\mathcal{M}$ can serve as a surrogate for the short geodesic arc length between them. As long as the step size is small and the curvature is not too large, this chord length is an accurate local approximation to the geodesic length.

The following proposition upgrades the local statement to an end-to-end guarantee: on any open set $U$ where $g|_U$ is a diffeomorphism onto its image, the geodesic distance between endpoints computed in latent space $(\mathcal{X}, \mathrm{g})$ exactly matches the geodesic distance between their images on the generated manifold $\mathcal{M}$. Consequently, shortest paths, interpolation costs, and energy minima are identical whether you pose the problem in $\mathcal{X}$ with the pullback metric or directly on $\mathcal{M}$ with the induced metric.

**Proposition 3.3** (Distance equality between $(\mathcal{X}, \mathbf{g})$ and $(\mathcal{M}, d_{\mathcal{M}})$)**.** *Endow $\mathcal{X}$ with the pullback Riemannian metric $\mathbf{g}_x(u, v) := \langle J_g(x)u, J_g(x)v \rangle$ and $\mathcal{M}_U$ with the Euclidean induced metric (hence geodesic distance $d_{\mathcal{M}}$). Then for all $x_1, x_2 \in U$,*

$$d_{\mathbf{g}}(x_1, x_2) \;=\; d_{\mathcal{M}}\big(g(x_1), g(x_2)\big),$$

*where $d_{\mathbf{g}}$ is the geodesic distance of $(\mathcal{X}, \mathbf{g})$ and $d_{\mathcal{M}}$ is the geodesic distance on $(\mathcal{M}_U, \langle \cdot, \cdot \rangle_2)$.*

*Proof.* For any absolutely continuous curve $\gamma : [0, 1] \to U$ we have, by definition of the pullback metric,

$$\big\| \dot{\gamma}(t) \big\|_{g, \gamma(t)} = \sqrt{\dot{\gamma}(t)^\top J_g(\gamma(t))^\top J_g(\gamma(t))\, \dot{\gamma}(t)} = \big\| \tfrac{d}{dt}\big(g \circ \gamma\big)(t) \big\|_2.$$

Since length is preserved pointwise and hence after integration, we have $L_{(\mathcal{X}, \mathbf{g})}(\gamma) = L_{(\mathcal{M}_U, \|\cdot\|_2)}(g \circ \gamma)$. Taking infimum over all $\gamma$ with $\gamma(0) = x_1, \gamma(1) = x_2$ yields

$$d_{\mathbf{g}}(x_1, x_2) \;\geq\; d_{\mathcal{M}}(g(x_1), g(x_2)),$$

because any competitor in $\mathcal{X}$ maps to a competitor in $\mathcal{M}_U$ with the same length.

Conversely, since $g|_U$ is a diffeomorphism, every absolutely continuous $\eta : [0, 1] \to \mathcal{M}_U$ with $\eta(0) = g(x_1), \eta(1) = g(x_2)$ lifts uniquely to $\gamma = (g|_U)^{-1} \circ \eta$ in $U$, and the same length identity shows $L_{(\mathcal{X}, \mathbf{g})}(\gamma) = L_{(\mathcal{M}_U, \|\cdot\|_2)}(\eta)$. Taking infimum over such $\eta$ gives the reverse inequality and hence equality of distances. $\square$

### 3.3 GEODESIC-GUIDED STYLE INTERPOLATION

In the context of style transfer, we aim to find an optimal path between content and style representations. Let us consider two points $x_c, x_s \in \mathcal{X}$ representing the content and style images in the coordinate space $\mathcal{X}$. These points can be connected by a continuous curve $\gamma : [0, 1] \to \mathcal{X}$ with $\gamma(0) = x_c$ and $\gamma(1) = x_s$. Following Section B, when this curve is mapped through the generative model $g$, it creates a trajectory $\gamma_g : [0, 1] \to \mathcal{M}$ on the Riemannian manifold $\mathcal{M}$, where $\gamma_g = g \circ \gamma$. Each point along this manifold curve represents a unique blend of content and style characteristics, enabling controlled style transfer.

To begin finding this optimal path, we start with a simple linear interpolation $x_m = \delta x_s + (1 - \delta)x_c$, where $\delta \in [0, 1]$ controls the mixing ratio between content and style. While this linear interpolation provides an initial point in the coordinate space $\mathcal{X}$, our goal is to refine it into a more geometrically meaningful path that better preserves both content and style characteristics when mapped onto the manifold $\mathcal{M}$ through the generative model $g$.

The curve $\gamma$ follows a direction from the content point $x_c$ to the style point $x_s$, and this directional path is preserved when mapped to the corresponding curve on the Riemannian manifold $\mathcal{M}$ through the generative model $g$. By performing forward finite difference, we can obtain the approximate velocity of the curve $\gamma$ at point $x_m$:

$$v_m = (g(x_s) - g(x_m))/(1 - \delta). \tag{1}$$

To quantify the optimality of this path, we derive a discrete version of the energy function from Equation 12 (refer to Appendix B).

$$E \approx \alpha(\|g(x_m) - g(x_c)\|^2 + \|g(x_s) - g(x_m)\|^2), \tag{2}$$

where $\alpha$ is a normalization factor that accounts for the relative distances between points. By our theoretical results in Section 3.2, the geodesic distance in space $\mathcal{X}$ is equivalent to the $\|\cdot\|_2$ norm in space $\mathcal{M}$ by the mapping $g$, which aligns with the insight in Equation 2.

To get the geodesic path with fixed start point and end point as $x_c$ and $x_s$, we need to minimize the discrete energy function. One way is to perform gradient descent on $x_m$, the gradient on the point $x_m$ is:

$$\nabla_{x_m} E = -\alpha J_g^\top(x_m)(g(x_s) - 2g(x_m) + g(x_c)). \tag{3}$$

The gradient in Equation 3 consists of two key components: a finite-difference second derivative on the manifold $\mathcal{M}$ and the Jacobian of $g$ derived through the chain rule. The finite-difference term in $\mathcal{M}$ space contains both tangential and normal components relative to the manifold's tangent space. The Jacobian $J_g$ serves to project out the normal component, effectively mapping the gradient from the manifold $\mathcal{M}$ back to the tangent space of $\mathcal{X}$, as illustrated in Figure 1. Through this projection mechanism, we can perform gradient descent directly on $x_m$ in the coordinate space $\mathcal{X}$ to find the geodesic path while respecting the manifold's geometric structure.

## 3.4 EFFICIENT JACOBIAN APPROXIMATION

However, computing the Jacobian of a generative model is computationally expensive, particularly for diffusion-based models. To address this challenge, we approximate the singular value decomposition (SVD) of the Jacobian using power iteration Haas et al. (2024) over the matrix $J_g^\top J_g$, whose eigenvectors correspond to the right singular vectors of $J_g$.

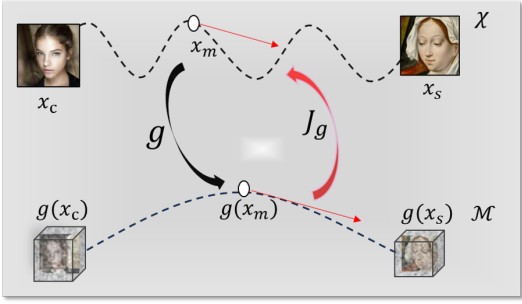

Figure 1: Geometric interpretation of the gradient computation: the generative model $g$ maps points from coordinate space $\mathcal{X}$ to the manifold $\mathcal{M}$, while the Jacobian $J_g$ projects the manifold gradient onto the tangent space of $\mathcal{X}$.

In our implementation, we focus on diffusion-based generative models that employ U-Net architectures as noise estimators. The U-Net's skip-connection structure creates a direct dependency between the network's output and both its input and hidden states. Consequently, when considering the hidden space $\mathcal{H}$, directions that induce changes in the network's predictions correspond directly to directions in the input space $\mathcal{X}$, enabling efficient computation of the Jacobian approximation. Consider the dependency on the hidden variables, the SVD of the Jacobian matrix can be written as:

$$J_g = \frac{\partial g_u(h_m)}{\partial h_m} = U\Sigma V^\top, h_m = g_d(x_m), \tag{4}$$

where $g_u, g_d$ denotes the up module and the down module of the U-Net network respectively, the right singular vectors corresponding to the largest singular values. We begin with a random vector $v$, the product $J_g^\top$ can be computed as:

$$J_g^\top J_g v = \frac{\partial}{\partial h}(g_u(h_m), J_g v) \tag{5}$$

and

$$J_g v = \frac{\partial}{\partial a} g_u(h_m + av)|_{a=0}. \tag{6}$$

Through Equation 5, the singular vector $v$ of the Jacobian matrix can be calculated. And thus the gradient with respect to $x_m$ can be modified as:

$$\eta_m = -\alpha v(g(x_s) - 2g(x_m) + g(x_c)), \tag{7}$$

then by applying gradient descent to point $x_m$ as $x_m \leftarrow x_m - \lambda \eta_m$, we can guide $g(x_m)$ along the geodesic path, enabling controlled style interpolation.

## 3.5 STYLE INJECTION CONTROL

Interpolating latent codes as in Section 3.3 can degrade content details. To better preserve content, prior work Chung et al. (2024) injects style via self-attention (SA) layers, using query features from the content image and key/value features from the style image. However, since query features largely determine spatial structure, using only the content query weakens the applied style.

We propose to find an optimal interpolation between the content query ($Q_c$) and style query ($Q_s$). A naive linear interpolation, $Q_m = (1 - \beta)Q_c + \beta Q_s$, often produces ghosting artifacts. Instead, we find a geodesic path in the query feature space by adapting the energy minimization from Section 3.3. We initialize $Q_m$ with linear interpolation and then minimize the energy:

$$\nabla_{Q_m} E = -k J_f^\top (f(Q_c) - 2f(Q_m) + f(Q_s)), \tag{8}$$

where $f$ is the attention module and $k = \frac{1}{\beta(1-\beta)}$. The key and value features are held fixed.

We approximate the Jacobian $J_f$ using power iteration as before. The vector-Jacobian products are computed via:

$$J_f^\top J_f v = \frac{\partial}{\partial Q_m}(f(Q_m), J_f v) \tag{9}$$

and

$$J_f v = \frac{\partial}{\partial a} f(Q_m + av)|_{a=0}, \tag{10}$$

Unlike the U-Net Jacobian in Equation 5, this computation is direct. By performing gradient descent on $Q_m$, we obtain modified query features that preserve content structure while faithfully representing the desired style.

## 4 EXPERIMENT

### 4.1 EXPERIMENT SETTING

**Implementation Details:** We use the official pretrained Stable Diffusion models, and the sample step is set to 50, the guidance scale is set to 3.0. The parameter $\delta$ is set to 0.9, $\alpha$ is set to 5, $\beta$ is set to 0.25 in our method. The experiment is conducted on a single NVIDIA H800 GPU.

**Evaluation Metric:** We quantitatively evaluate our method using ArtFID Wright & Ommer (2022) to measure style alignment, LPIPS Zhang et al. (2018) and Content Loss Li et al. (2017) to assess content preservation, and an identity preservation metric (ID) Jeong et al. (2024) based on ArcFace Deng et al. (2019) embeddings.

**Dataset:** Our experiments utilize three datasets: CelebA Liu et al. (2015) for real-world faces, MetFace Karras et al. (2020a) and WikiArt Tan et al. (2018) for artistic styles. Following standard protocol Chung et al. (2024); Deng et al. (2022), we randomly sample 20 content images from CelebA and 40 style reference images from each artistic dataset. For WikiArt, we use images from WikiArt-Face [1] as the reference style facial images, reprocessing the images by cropping facial regions with 0.5 expansion ratio and resizing to $512 \times 512$. And for MetFace, we just random select 40 images from MetFace [2] without any reprocessing.

**Baselines:** To evaluate the effectiveness of our proposed StyleBrush, we compare against several state-of-the-art style transfer approaches. We select representative methods from two categories: (1) diffusion-based methods including StyleSSP Xu et al. (2025), StyleID Chung et al. (2024), DiffuseIT Kwon & Ye (2022b), InST Zhang et al. (2023b), and DiffStyle Jeong et al. (2024); and (2) conventional methods including AesPA-Net Hong et al. (2023), CAST Zhang et al. (2022), StyTR$^2$ Deng et al. (2022), and ArtFlow An et al. (2021).

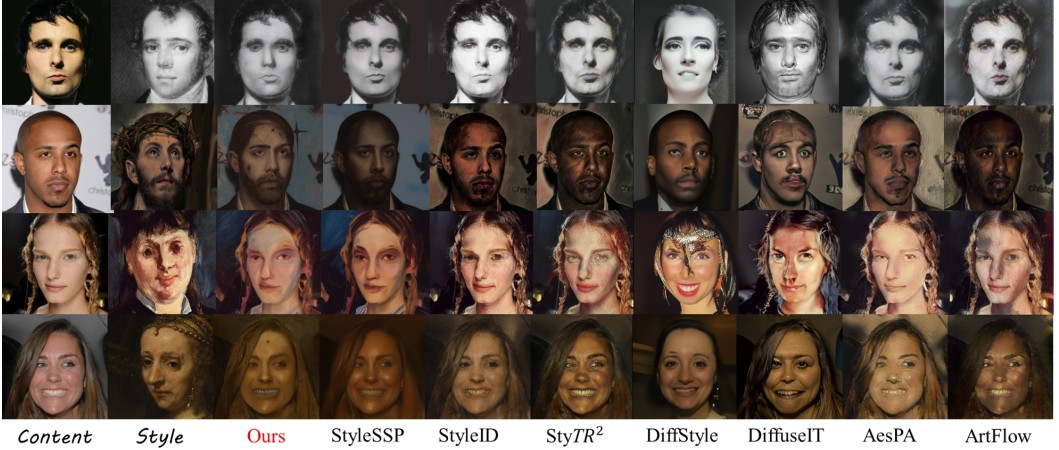

Figure 2: Visual comparison of style transfer results. Our StyleBrush better preserves facial identity while effectively transferring the artistic style compared to baseline methods.

---

[1] https://huggingface.co/datasets/asahi417/wikiart-face
[2] https://github.com/NVlabs/metfaces-dataset

## 4.2 MAIN RESULTS

We evaluate our StyleBrush against state-of-the-art baselines using WikiArt (diverse artistic paintings) and MetFace (artistic portraits) as style references, and CelebA for content images. Comparisons include both CNN-based methods and diffusion-based approaches. For fair comparison, all diffusion-based methods use Stable Diffusion v1.4 as the base model. To distinguish our method with and without the additional control, we denote our method with and without the additional control as StyleBrush and StyleBrush *, respectively.

Table 1: The style transfer results with using WikiArt and MetFace datasets as style reference.

| Method | WikiArt | | | | MetFace | | | |
|---|---|---|---|---|---|---|---|---|
| | ArtFID ($\downarrow$) | LPIPS ($\downarrow$) | ID ($\downarrow$) | CS ($\uparrow$) | ArtFID ($\downarrow$) | LPIPS ($\downarrow$) | ID ($\downarrow$) | CS ($\uparrow$) |
| AesPA-Net Hong et al. (2023) | 28.3402 | 0.4076 | 0.2696 | 0.4254 | 29.1623 | 0.4853 | 0.2968 | 0.3948 |
| ArtFlow An et al. (2021) | **22.9778** | 0.4933 | 0.2555 | 0.3215 | 30.2701 | 0.4717 | 0.2484 | 0.3932 |
| CAST Zhang et al. (2022) | 33.0048 | 0.3856 | 0.2832 | 0.4064 | 28.6567 | 0.4109 | 0.2877 | 0.4293 |
| StyTR$^2$ Deng et al. (2022) | 24.2880 | 0.4068 | 0.1760 | 0.5181 | 26.6994 | 0.4104 | 0.1599 | 0.5503 |
| InST Zhang et al. (2023b) | 29.8916 | 0.5975 | 0.8075 | 0.1862 | 33.4454 | 0.6084 | 0.8432 | 0.2447 |
| DiffStyle Jeong et al. (2024) | 43.5315 | 0.6084 | 0.7988 | 0.3220 | 46.6953 | 0.6128 | 0.7824 | 0.3195 |
| DiffuseIT Kwon & Ye (2022b) | 40.2349 | 0.5658 | 0.6619 | 0.2845 | 37.3332 | 0.5540 | 0.6542 | 0.2912 |
| StyleID Chung et al. (2024) | 24.0963 | 0.3785 | 0.1618 | 0.4493 | 26.6917 | 0.5283 | 0.1646 | 0.4283 |
| StyleBrush (Ours) | 24.4906 | **0.2616** | **0.1237** | **0.5531** | 26.6203 | **0.3118** | **0.1424** | **0.5618** |
| Diffusion-based methods with additional Control[1] | | | | | | | | |
| StyleSSP Xu et al. (2025) | 36.3980 | 0.4535 | 0.1707 | 0.4281 | 35.6287 | **0.4079** | 0.1658 | **0.5322** |
| StyleBrush * (Ours) | **35.1855** | **0.4523** | **0.1433** | **0.4572** | **31.2170** | 0.5228 | **0.1441** | 0.4454 |

[1] Include both ControlNet Zhang et al. (2023a) and IP adapter Ye et al. (2023).

As shown in Table 1, our StyleBrush demonstrates superior performance compared to existing CNN-based and diffusion-based approaches across multiple quantitative metrics. Specifically, on the WikiArt dataset, our approach achieves state-of-the-art results in LPIPS (0.2616), identity preservation (0.1237), and content similarity (0.5531). The performance advantages are further validated on the MetFace dataset, where our StyleBrush exhibits leading scores across all evaluation metrics, including ArtFID (26.6203), LPIPS (0.3118), and identity preservation (0.1424).

While StyleID and StyTR$^2$ show competitive identity preservation, they exhibit suboptimal style transfer capabilities. Other diffusion-based methods yield higher ArtFID and LPIPS scores, indicating limitations in both stylization and content preservation. Visual comparisons (Figure 2) further validate our method's superior performance in balancing identity preservation and style transfer.

## 4.3 ABLATION STUDY

We perform ablation studies to understand the contribution of each component of our StyleBrush. We remove the style interpolation initial (see Section 3.3) and the style injection control (see Section 3.5) to see the performance of our StyleBrush. The results are shown in Table 2 and Figure 3.

The ablation study results in Table 2 and Figure 3 analyze our StyleBrush's key components: style interpolation initialization and style injection control. Removing style interpolation initialization maintains similar Art-FID but degrades perceptual quality, with LPIPS increasing from 0.3118 to 0.3647 and ID worsening from 0.1424

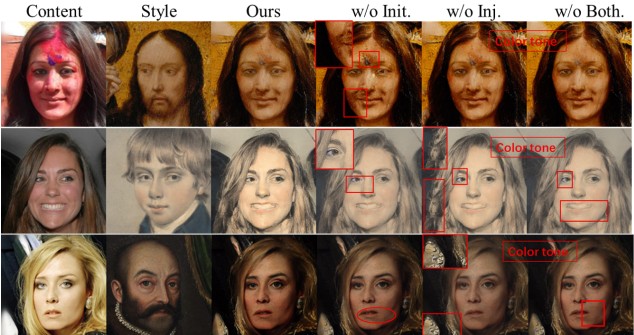

Figure 3: Ablation study visualization. From left to right: content image, style image, full results of StyleBrush, results without style interpolation initialization, results without style injection control, results without both components.

to 0.1250. Figure 3 shows this degradation visually, particularly in facial features where stylized outputs lack coherent adaptation of eye shapes and skin textures compared to our full StyleBrush.

Analysis of the style injection control ablation reveals that its removal, while marginally improving ArtFID, significantly degrades other performance metrics. Specifically, we observe a 23.8% deterioration in LPIPS (0.3118 to 0.3861) and 15.7% decrease in content similarity (0.5618 to 0.4735), in-

Table 2: Ablation study results comparing our full method against variants with key components removed. Experiments are evaluated with using the MetFace dataset as style images.

| Method | ArtFID (↓) | LPIPS (↓) | ID (↓) | CS (↑) |
|---|---|---|---|---|
| w/o Sty. Interp. Init. | 26.4807 | 0.3647 | **0.1250** | 0.4933 |
| w/o Sty. Inject Control | **26.3421** | 0.3861 | 0.1470 | 0.4735 |
| w/o both | 26.7091 | 0.3877 | 0.1473 | 0.4715 |
| StyleBrush | 26.6203 | **0.3118** | 0.1424 | **0.5618** |

dicating compromised content preservation. The ablation of both components yields the most severe degradation, empirically validating their synergistic relationship. Our complete StyleBrush demonstrates optimal performance across metrics, with qualitative results in Figure 3 confirming superior style-content integration while maintaining facial fidelity.

## 4.4 Linear Style Interpolation

In Section 3.5, we demonstrate our StyleBrush's capability for injecting the style information through a continuous interpolation parameter $\beta \in [0, 1]$. As evidenced by visual results and quantitative measurements (refer to Figure 4), our approach achieves smooth transitions from content to style while maintaining semantic consistency. The gradual change in facial features follows a linear trajectory as $\beta$ increases, with lower values ($\beta = 0.2, 0.4$) preserving content fidelity while incorporating subtle stylistic elements, and higher values ($\beta = 0.6, 0.8$) intensifying artistic characteristics while maintaining facial features. The stylized images maintain coherent content information due to the

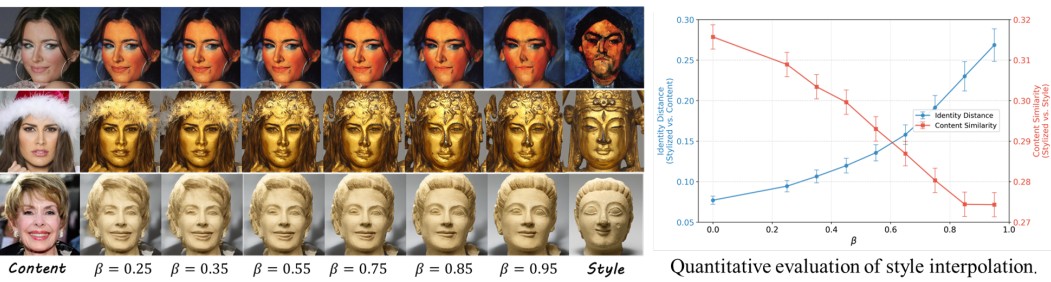

Content    $\beta = 0.25$    $\beta = 0.35$    $\beta = 0.55$    $\beta = 0.75$    $\beta = 0.85$    $\beta = 0.95$    Style      Quantitative evaluation of style interpolation.

Figure 4: Style interpolation results showing smooth transitions between different artistic styles.

initial linear interpolation that fuses the content and the style information (with $\delta = 0.9$), which demonstrates the effectiveness of our geodesic-based style interpolation mechanism in preserving semantic structure while enabling controlled style transfer. Quantitatively, this effectiveness is validated by the monotonic progression of identity loss values across interpolation steps. These results empirically demonstrate our StyleBrush's ability to achieve precise, continuous control over the content-style trade-off while maintaining semantic consistency throughout the interpolation process.

## 4.5 Geometric Properties of Interpolation Paths

To validate that geodesic paths provide superior interpolation compared to simple linear paths, we analyze the geometric properties of different interpolation strategies in the diffusion latent space. Our theoretical framework relies on the local isometry property, which requires well-behaved manifold geometry. We empirically verify this through curvature analysis and comparison with alternative interpolation schemes. We compute Ricci curvature along two types of paths: (1) linear interpolation $z(t) = (1 - t)z_c + t z_s$ in latent space, (2) geodesic paths computed via our energy minimization framework, and (3) randomly perturbed paths.

As shown in Figure 5, geodesics consistently exhibit Ricci curvature values of 0.2-0.3 across diverse style pairs. This moderate curvature indicates: (1) the latent manifold is mildly curved, as expected for nonlinear generative mappings, and (2) the curvature magnitude remains small, validating our local isometry assumption and ensuring stable geodesic computation. Linear interpolation yields Ricci $\approx 0$ because it assumes Euclidean geometry by construction, not because the manifold is actually flat. In contrast, geodesic paths reveal the intrinsic geometry. This contrast explains why geodesics outperform linear paths: linear interpolation's flat-space assumption causes traversal

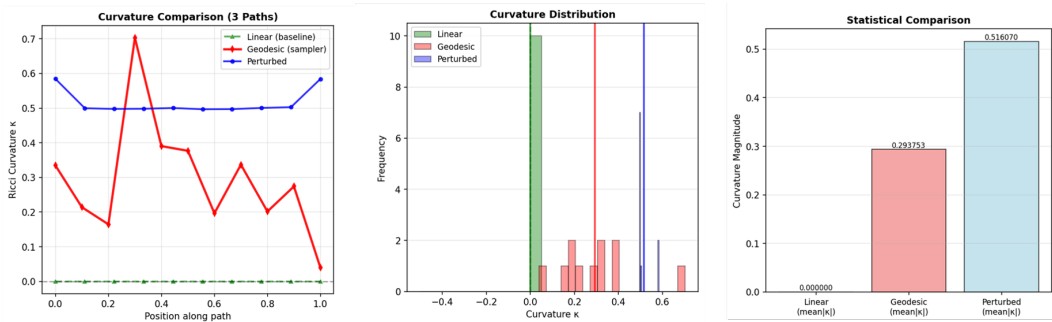

Figure 5: Comparison of interpolation path geometries. From left to right, the curvature of different paths, the curvature distribution of different paths, and the curvature magnitude of different paths.

through geometrically inconsistent regions, producing artifacts, while geodesics respect the manifold's natural curvature, maintaining semantic consistency.

### 4.6 METHOD HYPERPARAMETER ANALYSIS

We analyze the influence of the interpolation parameters $\delta$ and $\beta$ on style transfer performance. As shown in Figure 6, $\delta$ controls the stability of the geodesic path optimization. Larger values lead to more stable geodesic trajectories, while smaller values introduce instability. We empirically set $\delta = 0.9$ to ensure stability without excessive style injection. The parameter $\beta$ balances style strength and content preservation, as illustrated in Figure 7. ArtFID scores initially improve with increasing $\beta$ but then degrade as the stylized image diverges from the original content. We find $\beta = 0.25$ provides the best trade-off between style transfer quality and content fidelity.

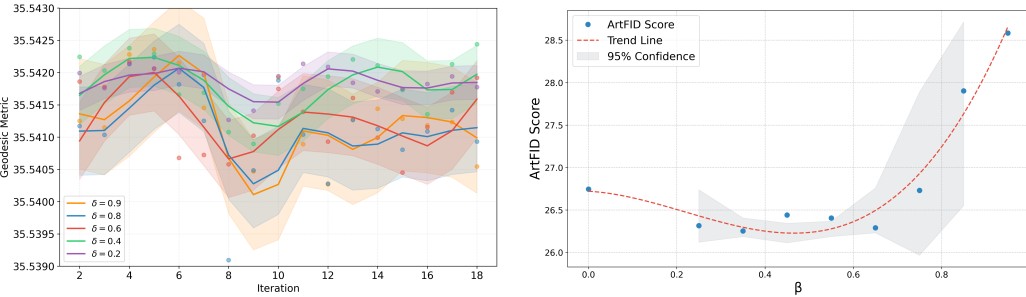

Figure 6: The effect of $\delta$ on identity preservation and content similarity.

Figure 7: The effect of $\beta$ on style transfer performance.

## 5 CONCLUSION

We presented StyleBrush, a novel facial stylization framework grounded in Riemannian geometry that provides a principled solution to the challenge of controlled style transfer while preserving identity. Our core theoretical contribution is the formulation of style interpolation as a geodesic path-finding problem on a latent manifold defined by the generative model. By leveraging the pullback Riemannian metric, we establish a local isometry between the latent space and the generated manifold, a key result that theoretically validates minimizing a discrete path energy in the embedding space as a proxy for finding a true geodesic, enabling smooth and semantically consistent content-style fusion. Furthermore, we extend this geometric perspective by applying the same geodesic optimization to the query features within the model's self-attention layers, yielding a dynamic style injection mechanism that precisely modulates stylization while maintaining structural integrity. Our experimental results, which demonstrate state-of-the-art performance, validate the practical efficacy of our geometric framework. The framework's generality invites future theoretical inquiry into global manifold properties, connections to optimal transport theory, and the development of convergence guarantees for the proposed optimization.

## 6 ETHICS STATEMENT

Our research utilizes publicly available datasets, including CelebA, WikiArt, and MetFace, for development and evaluation. We acknowledge that image datasets containing human faces may carry inherent demographic biases, which could potentially be reflected in the model's outputs. This work is intended solely for academic research and creative applications. We recognize the potential for misuse of facial generation technologies in creating deceptive or harmful content. We strongly condemn any such malicious applications and hope our work contributes positively to the fields of computer graphics and artistic expression. No private user data was collected for this research, and we encourage the responsible use of this technology.

## 7 REPRODUCIBILITY STATEMENT

To ensure full reproducibility of our findings, we commit to releasing our source code and experimental configurations upon publication of this work. Our framework is implemented on top of standard, publicly available pre-trained Stable Diffusion models (v1.4, v1.5, v2.0, and v2.1). The datasets used in our experiments are CelebA, MetFace, and the WikiArt-Face subset. All crucial hyperparameters, including the guidance scale, sampling steps, and the parameters specific to our method ($\delta$, $\alpha$, $\beta$), are detailed in the main text. The experiments were conducted on a single NVIDIA H800 GPU, and our evaluation protocol is described in detail to facilitate faithful replication of our results.

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

## A  USE OF LLMS

During the preparation of this manuscript, we utilized a large language model (LLM) as a writing and editing assistant. The LLM's role was primarily to improve the clarity, conciseness, and academic tone of the text. Specific tasks included paraphrasing sentences, ensuring grammatical correctness, and maintaining a consistent narrative voice across sections such as the abstract and conclusion. All technical and theoretical content, including the core methodology, experimental design, and analysis of results, was exclusively developed by the human authors. The final manuscript was thoroughly reviewed and edited by the authors to ensure the accuracy and integrity of all claims.

## B  GEODESIC PATH COMPUTATION

The latent space of a generative model $g$ can be formally characterized as a Riemannian manifold Shao et al. (2018), endowed with a smooth structure and a metric tensor. For any two points in the coordinate space $\mathcal{X}$, their mappings through $g$ onto this manifold are connected by a unique geodesic path, which represents the locally length-minimizing curve on the Riemannian manifold. Consider two points $x_1, x_2 \in \mathcal{X}$ in the coordinate space. Let $\gamma : [0, 1] \to \mathcal{X}$ be a smooth path such that $\gamma(0) = x_1$ and $\gamma(1) = x_2$. The composition map $g \circ \gamma : [0, 1] \to \mathcal{M}$ induces a unique curve on the Riemannian manifold $\mathcal{M}$, where $g \circ \gamma(t) \in \mathcal{M}$ for all $t \in [0, 1]$. This construction preserves the differential structure of $\gamma$ under the mapping $g$, establishing a diffeomorphism between the path in coordinate space and its image on $\mathcal{M}$. To characterize the geometry relationship between the latent space and the manifold, we have the following definition.

**Definition B.1** (Pullback metric **g** on $\mathcal{X}$ induced by the generator). Let $g : \mathcal{X} = \mathbb{R}^n \to \mathcal{M} \subset \mathbb{R}^p$ be a continuously differentiable map with Jacobian $J_g(x) \in \mathbb{R}^{p \times n}$. For each $x, u, v \in \mathcal{X}$, define

$$\mathbf{g}_x(u, v) \ := \ \big\langle J_g(x)\, u, \ J_g(x)\, v \big\rangle_{\mathbb{R}^p}.$$

In local coordinates, the matrix representation of $\mathbf{g}_x$ is

$$G(x) \ = \ J_g(x)^\top J_g(x),$$

so that for $u, v \in \mathbb{R}^n$,

$$\mathbf{g}_x(u, v) \ = \ u^\top G(x)\, v, \qquad \|u\|_{g,x}^2 \ = \ u^\top G(x)\, u \ = \ \big\| J_g(x)\, u \big\|_2^2.$$

If $J_g(x)$ has full column rank, then $\mathbf{g}_x$ is a Riemannian metric on $\mathcal{X}$.

Given this metric, we can formally define the arc length of a smooth path $\gamma$ through the Riemannian line integral:

$$L(\gamma) = \int_0^1 \sqrt{\frac{d\gamma(t)}{dt}^\top G(\gamma(t)) \frac{d\gamma(t)}{dt}}\, dt = \int_0^1 \sqrt{\mathbf{g}_{\gamma(t)}\Big(\frac{d\gamma(t)}{dt}, \frac{d\gamma(t)}{dt}\Big)}\, dt. \tag{11}$$

This integral measures the accumulated squared speed of the path under the pullback metric. The corresponding geodesic curve minimizes this arc length locally, which is equivalent to minimizing the energy function:

$$E(\gamma) = \frac{1}{2} \int_{x_1}^{x_2} \frac{d\gamma(t)}{dt}^\top G(\gamma(t)) \frac{d\gamma(t)}{dt}\, dt. \tag{12}$$

Using energy is standard because it is smooth and makes the calculus of variations straightforward. Then we express the Christoffel symbols of the metric $G$, denoted as $\Gamma_{jk}^i$:

$$\Gamma_{jk}^i = \frac{1}{2} G_{il}^{-1} \Big( \frac{\partial G_{lj}}{\partial x^k} + \frac{\partial G_{lk}}{\partial x^j} - \frac{\partial G_{jk}}{\partial x^l} \Big), \tag{13}$$

where $i, j, k$ are indices of the coordinate space. Operationally, they tell us how to differentiate tangent vectors along the manifold. By taking a variation of the energy function in Equation 12 and applying the calculus of variations, we derive the Euler-Lagrange equation:

$$\frac{d^2\gamma^i}{dt^2} = -\Gamma_{jk}^i \frac{d\gamma^j}{dt} \frac{d\gamma^k}{dt}. \tag{14}$$

This second-order ordinary differential equation is the stationary condition of the energy functional. The geodesic path $\gamma$ can then be computed through numerical integration of Equation 14.

Then we derive the approximation to $E$ in Equation (2). Let $y(t) := g(\gamma(t)) \in \mathcal{M}$ and recall the continuous energy on $(\mathcal{X}, \mathbf{g})$ equals the kinetic energy of $y$ on $\mathcal{M}$:

$$E(\gamma) = \frac{1}{2} \int_0^1 \left\| \dot{y}(t) \right\|_2^2 dt \quad \text{(a.e. in } t \text{, since } g \in C^1, \gamma \in AC\text{)}.$$

Fix $\delta \in (0,1)$ and split $[0,1]$ into two subintervals $[0,\delta]$ and $[\delta,1]$. Write $y_c := g(x_c) = y(0)$, $y_m := g(x_m) = y(\delta)$, $y_s := g(x_s) = y(1)$. Then

$$E(\gamma) = \frac{1}{2} \left( \int_0^\delta \|\dot{y}(t)\|_2^2 \, dt + \int_\delta^1 \|\dot{y}(t)\|_2^2 \, dt \right).$$

Assume on each subinterval the mapped curve $y(t)$ is well approximated by a straight segment with constant speed. Using forward/backward finite differences,

$$\dot{y}(t) \approx \begin{cases} \dfrac{y_m - y_c}{\delta}, & t \in [0,\delta], \\[2mm] \dfrac{y_s - y_m}{1-\delta}, & t \in [\delta,1]. \end{cases}$$

Plugging into the energy gives the standard two-segment discretization:

$$E(\gamma) \approx \frac{1}{2} \left( \delta \left\| \frac{y_m - y_c}{\delta} \right\|_2^2 + (1-\delta) \left\| \frac{y_s - y_m}{1-\delta} \right\|_2^2 \right)$$

$$= \frac{1}{2} \left( \frac{1}{\delta} \|y_m - y_c\|_2^2 + \frac{1}{1-\delta} \|y_s - y_m\|_2^2 \right). \tag{A.1}$$

Bringing (A.1) over the common denominator $2\delta(1-\delta)$ yields

$$E(\gamma) = \frac{1}{2\delta(1-\delta)} \Big( (1-\delta)\|y_m - y_c\|_2^2 + \delta\|y_s - y_m\|_2^2 \Big). \tag{A.2}$$

Defining $\alpha := \frac{1}{2\delta(1-\delta)}$, we obtain the compact form

$$E(\gamma) \approx \alpha \Big( (1-\delta) \, \|g(x_m) - g(x_c)\|_2^2 + \delta \, \|g(x_s) - g(x_m)\|_2^2 \Big). \tag{A.3}$$

When the midpoint $x_m$ corresponds to equal time split ($\delta = \frac{1}{2}$), (A.3) reduces to

$$E(\gamma) \approx \|g(x_m) - g(x_c)\|_2^2 + \|g(x_s) - g(x_m)\|_2^2,$$

i.e., the unweighted two-chord sum (here $\alpha = 1$).

## C   BACKGROUND

### C.1   ATTENTION-BASED STYLE INJECTION

Attention-based style injection Chung et al. (2024) is a method used to incorporate style information into the latent representation of content images, offering a flexible and precise control over style transfer. This approach combines DDIM inversion with Self-Attention (SA) to effectively transfer style while maintaining content structure.

In this framework, the latents for both the content and style images are first obtained through DDIM inversion. Given a predefined set of timesteps $t \in \{0, \ldots, T\}$, the content latent $\mathbf{z}_c^0$ and style latent $\mathbf{z}_s^0$ undergo a deterministic forward process, progressively transforming into Gaussian noise distributions at timestep $t = T$. During this inversion process, we extract and store the query features $Q_t^c$ from the content image and the key-value pairs $(K_t^s, V_t^s)$ from the style image at each timestep, which encode the structural and stylistic information respectively.

The stylization process begins by initializing the stylized latent noise $\mathbf{z}_{cs}^T$ with the content latent noise $\mathbf{z}_c^T$. Style transfer is then achieved through attention feature modulation, where the collected style features $(K_t^s, V_t^s)$ are injected into the Self-Attention layer, replacing the original content features $(K_t^{cs}, V_t^{cs})$. This feature substitution enables precise style integration while preserving the content's structural integrity in the latent representation.

The synergy between DDIM inversion and attention-based feature modulation provides a theoretically grounded approach for style transfer, offering fine-grained control over the stylization process while maintaining the mathematical properties of both the diffusion and attention mechanisms.

## C.2 DDIM INVERSION

DDIM inversion is a forward process that progressively adds estimated noise to an image from timestep 0 to T, enabling the reconstruction of the noise schedule that would generate the image. Unlike the sampling process which denoises from T to 0, DDIM inversion follows the forward path to obtain the specific noise that characterizes the input image.

Given a clean image $\mathbf{x}_0$, DDIM inversion follows a deterministic forward process to progressively add noise using the learned noise predictor $\epsilon_\theta$. For timestep $t$, the forward update rule is defined as:

$$\mathbf{x}_{t+1} = \sqrt{\bar{\alpha}_{t+1}}\mathbf{x}_0 + \sqrt{1 - \bar{\alpha}_{t+1}}\epsilon_t$$

where $\bar{\alpha}_t = \prod_{s=1}^{t} \alpha_s$ represents the cumulative product of noise scheduling coefficients, and $\epsilon_t$ is obtained through the noise predictor $\epsilon_\theta(\mathbf{x}_t, t)$. This process iteratively continues from $t = 0$ to $T$, generating a sequence of increasingly noisy latents $\{\mathbf{x}_t\}_{t=1}^T$ that encode the image's characteristics in the model's latent space. The resulting noise schedule and latents can then be used for controlled generation through the reverse process.

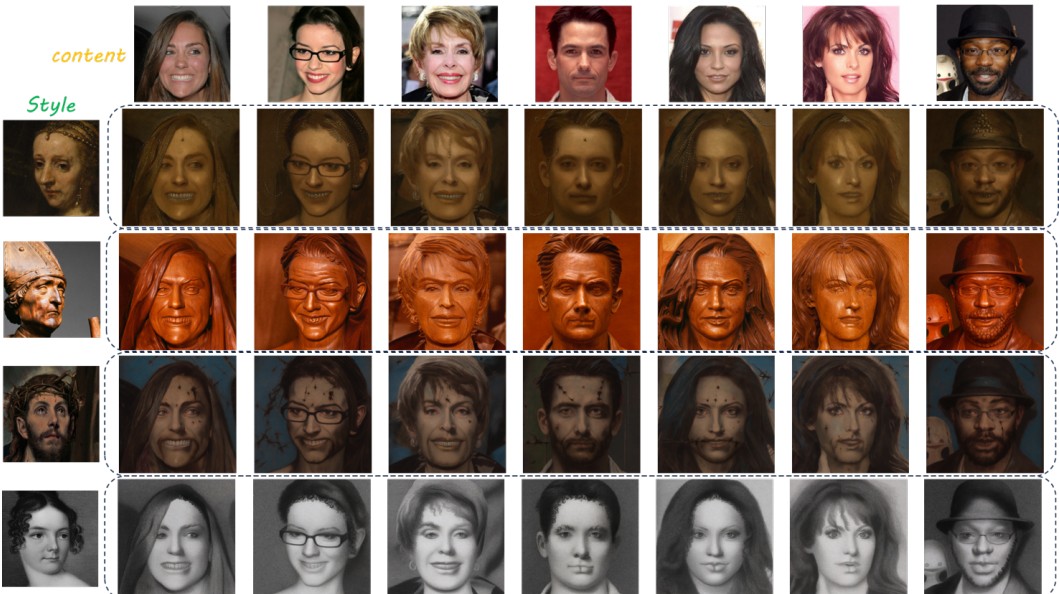

Figure 8: More Visualization Samples of the proposed StyleBrush *.

# D EXPERIMENT DETAILS

## D.1 IMPLEMENTATION DETAILS:

We use the official pretrained Stable Diffusion models, and the sample step is set to 50, the guidance scale is set to 3.0. The parameter $\delta$ is set to 0.9, $\alpha$ is set to 5, $\beta$ is set to 0.25 in our method. The experiment is conducted on a single NVIDIA H800 GPU.

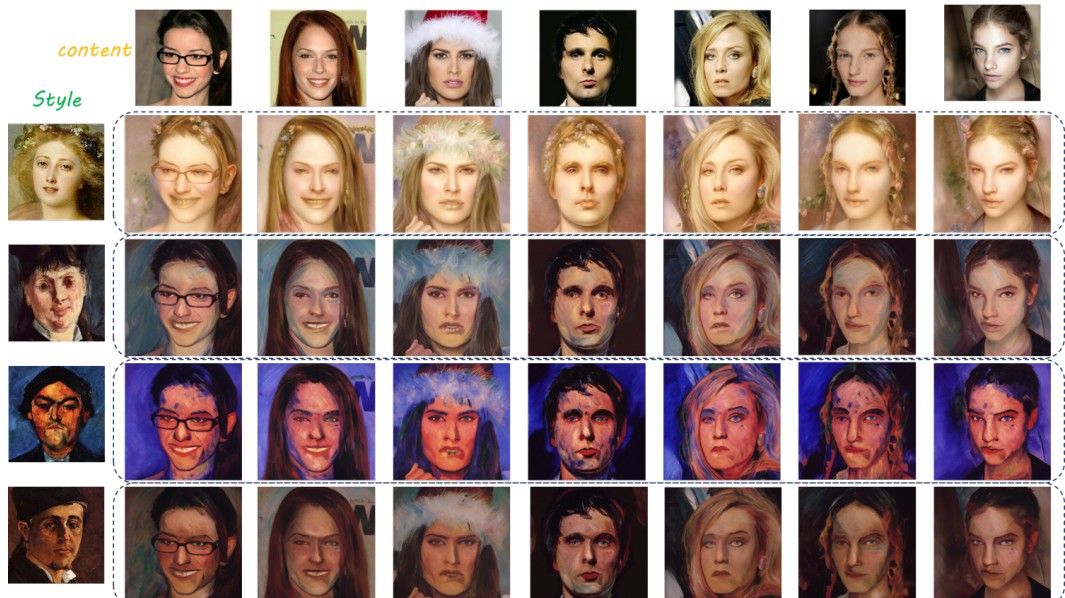

Figure 9: More Visualization Samples of the proposed StyleBrush *.

## D.2 IMPLEMENTATION OF BASELINE METHODS

For baseline methods, we use their official implementation and parameters.

- StyleSSP Xu et al. (2025): A semantic structure-preserving model for unsupervised portrait style transfer that focuses on maintaining the semantic layout of the face.

- StyleID Chung et al. (2024): A zero-shot portrait stylization method that uses a style-aware ID loss and injects style features into the self-attention layers of a diffusion model to preserve identity.

- DiffuseIT Kwon & Ye (2022b): A text-guided image translation framework based on diffusion models, adapted for style transfer by using textual descriptions of the style.

- InST Zhang et al. (2023b): An inversion-based style transfer method that first inverts a content image to a noise map and then modulates intermediate features of a diffusion model using the style reference during the reverse process.

- DiffStyle Jeong et al. (2024): A training-free, diffusion-based face stylization approach that blends noise predictions for content and style during the sampling process to achieve expressive results.

and four conventional style transfer methods:

- AesPA-Net Hong et al. (2023): An adversarial stylization method designed for aesthetically pleasing portrait animation, which can be applied to static image stylization.

- CAST Zhang et al. (2022): A transformer-based method that utilizes cross-attention mechanisms to align features between content and style images for effective style transfer.

- StyTR$^2$ Deng et al. (2022): An improved transformer-based framework that treats style transfer as a sequence-to-sequence task, using an encoder-decoder architecture to capture both global and local style patterns.

- ArtFlow An et al. (2021): A style transfer method based on reversible neural flows that provides a bias-free transformation, ensuring precise content preservation by learning invertible mappings between content and style feature distributions.

### D.3 COMPUTATION COST

We provide detailed analysis of computational requirements and compare our method with state-of-the-art baselines. The analysis demonstrates that our approach achieves superior quality-efficiency trade-off among training-free methods. **Timing Comparison.** Table 3 presents per-image inference time measurements for $512 \times 512$ images on a single NVIDIA H800 GPU. Our method processes one image in 9.71 seconds, which is dramatically faster than training-based diffusion methods: $23 \times$ faster than DiffuseIT (226.82s) and $7 \times$ faster than InjectFusion (68.24s). Among training-free methods, StyleID achieves faster inference (3.97s, $2.4 \times$ faster than ours), but delivers substantially lower quality across all metrics as discussed below.

Table 3: Computational comparison on $512 \times 512$ images (NVIDIA H800 GPU).

| Method | Inference Time (s) | Training Required | Speedup vs. Ours |
|---|---|---|---|
| DiffuseIT | 226.82 | Yes (per style) | $0.04 \times$ ($23 \times$ slower) |
| InjectFusion | 68.24 | Yes (per style) | $0.14 \times$ ($7 \times$ slower) |
| StyleID | 3.97 | No | $2.4 \times$ (faster) |
| Ours | 9.71 | No | $1.0 \times$ |

**Training Requirements.** A critical advantage of our approach is that it requires no training or fine-tuning. We use pretrained Stable Diffusion models directly without modification. In contrast, DiffuseIT and InjectFusion require training on style-specific datasets, which takes several hours on multiple GPUs and must be repeated for each new style domain. This training overhead severely limits flexibility for diverse style transfer applications. Our training-free nature enables immediate application to arbitrary new styles without any preparation beyond having the style reference image.

**Computational Efficiency.** Our geodesic optimization achieves efficiency through three key designs: (1) Hutchinson's trace estimator for Jacobian-vector products, using only 2-3 stochastic projections instead of materializing the full Jacobian matrix, (2) fast convergence in 5-10 iterations due to the smooth geometry of the learned manifold, with early stopping when energy changes drop below $10^{-4}$, and (3) activation checkpointing that recomputes forward passes during backpropagation rather than storing all intermediate activations. These optimizations make the geodesic computation practical while maintaining superior quality.

**Quality-Efficiency Trade-off.** The comparison with StyleID is particularly revealing. While StyleID is $2.4 \times$ faster, our method achieves substantially better quality (see Table 1): identity preservation (ID: 0.1237 vs. 0.1618 on WikiArt), perceptual distance (LPIPS: 0.2616 vs. 0.3785), and content similarity (CS: 0.5531 vs. 0.4493). The modest computational overhead of geodesic navigation is justified by significant quality improvements across all evaluation metrics. This represents the fundamental advantage of respecting manifold geometry: geodesic paths produce better results than simple interpolation schemes.

In summary, our method achieves the optimal balance for training-free style transfer: no domain-specific training required, practical inference times suitable for content creation workflows (approximately 370 images per hour), and superior quality compared to alternative training-free approaches. The 9.71-second processing time is entirely practical for professional applications while delivering state-of-the-art quality.

### D.4 PERFORMANCE ON DIFFERENT BASE MODEL

To evaluate the robustness and generalization capability of our StyleBrush, we conduct experiments using different versions of Stable Diffusion (SD) as the base model. We compare our approach with StyleID across SD v1.5, v2.0, and v2.1. Table 4 presents a comprehensive comparison of the performance metrics across these different base models.

The results in Table 4 demonstrate the consistent superiority of our StyleBrush across all SD versions. For the WikiArt dataset, our approach achieves better performance across all metrics, with notable improvements in LPIPS scores (reducing from 0.3878 to 0.2737 in SD v1.5) and content similarity (CS $> 0.55$). Similar improvements are observed with the MetFace dataset, where our StyleBrush exhibits superior identity preservation and style transfer quality. Notably, our StyleBrush maintains

Table 4: The style transfer results with using WikiArt and MetFace datasets as style reference.

| Base Model | Mwthod | WikiArt | | | | MetFace | | | |
|---|---|---|---|---|---|---|---|---|---|
| | | ArtFID (↓) | LPIPS (↓) | ID (↓) | CS (↑) | ArtFID (↓) | LPIPS (↓) | ID (↓) | CS (↑) |
| **SD v1.5** | StyleID | 20.6744 | 0.3878 | 0.1273 | 0.4679 | **25.3002** | 0.4044 | 0.0952 | 0.4742 |
| | StyleBrush | **20.2082** | **0.2737** | **0.1242** | **0.5502** | 26.5815 | **0.2436** | **0.0626** | **0.5941** |
| **SD v2.0** | StyleID | 20.8419 | 0.3850 | 0.1271 | 0.4715 | 25.3235 | 0.3992 | 0.0915 | 0.4817 |
| | StyleBrush | **20.0907** | **0.2508** | **0.1177** | **0.5758** | **24.5590** | **0.2580** | **0.0649** | **0.6033** |
| **SD v2.1** | StyleID | 25.3337 | 0.4200 | 0.1351 | 0.4244 | 27.9503 | 0.5190 | 0.1234 | 0.4077 |
| | StyleBrush | **20.8076** | **0.2527** | **0.1175** | **0.5783** | **26.7607** | **0.2601** | **0.0659** | **0.6016** |

consistent performance across all SD versions, demonstrating its robustness and generalizability regardless of the underlying diffusion model architecture.

## D.5 GENERAL STYLE TRANSFER

While our main paper focuses on facial style transfer, our method is fundamentally domain-agnostic. The geodesic framework relies purely on the learned manifold geometry of diffusion models without any face-specific priors or constraints. To validate this generality, we conduct comprehensive experiments across three diverse non-face domains: art-to-art transfer, object stylization, and scene stylization. To validate the domain-agnostic nature of our method, we conduct experiments on diverse non-facial content. We select content images from MS-COCO **?**, which provides natural images across various categories including objects, animals, and complex scenes. Style reference images are randomly sampled from WikiArt Tan et al. (2018), spanning diverse artistic movements. This setup tests whether our geodesic framework generalizes beyond facial images to arbitrary content-style pairs.

**Object Stylization.** We evaluate style transfer on objects with clear semantic structure, including animals (birds), and everyday objects (pizza). As shown in Figure 10 (top row), the geodesic paths preserve object structure and identity while applying artistic textures and color transformations. Object shapes remain intact while adopting the style's visual characteristics, validating that our manifold-based approach respects semantic boundaries beyond facial features.

**Scene Stylization.** We test on complex scenes from MS-COCO. Scene stylization is particularly challenging due to multiple objects, varying scales, and complex spatial relationships. Figure 10 (middle row) demonstrates that our method maintains spatial coherence and layout consistency while transferring style. The geodesic paths naturally preserve the scene's compositional structure—horizon lines, building arrangements, foreground-background relationships—while applying artistic transformations.

**Art-to-Art Transfer.** We test style transfer between artistic paintings without facial content, sampling painting pairs from WikiArt spanning diverse artistic movements. As shown in Figure 10 (bottom row), our method successfully captures and transfers artistic characteristics—brush stroke patterns, color palettes, and texture styles—while maintaining the content painting's compositional structure and spatial layout.

## D.6 CASE STUDY

**Using Non-Facial Art References** We evaluate our StyleBrush's capability to transfer styles from non-facial artistic references. As shown in Figure 11, our approach successfully extracts and applies artistic elements (color palettes, brush strokes, textures) from non-portrait artworks while preserving facial identity and human features. The results demonstrate that our StyleBrush can effectively handle diverse artistic references beyond portraits, from abstract paintings to landscapes, while maintaining content fidelity.

**Artistic style to artistic style** Our StyleBrush can effectively transfer artistic styles between different artistic images while preserving the content and identity of the source image. Figure 12 shows an example where we transfer the style from one artistic portrait to another. The results demonstrate that our StyleBrush successfully captures and transfers the distinctive artistic characteristics while maintaining the facial features and overall composition of the content image. This capability is

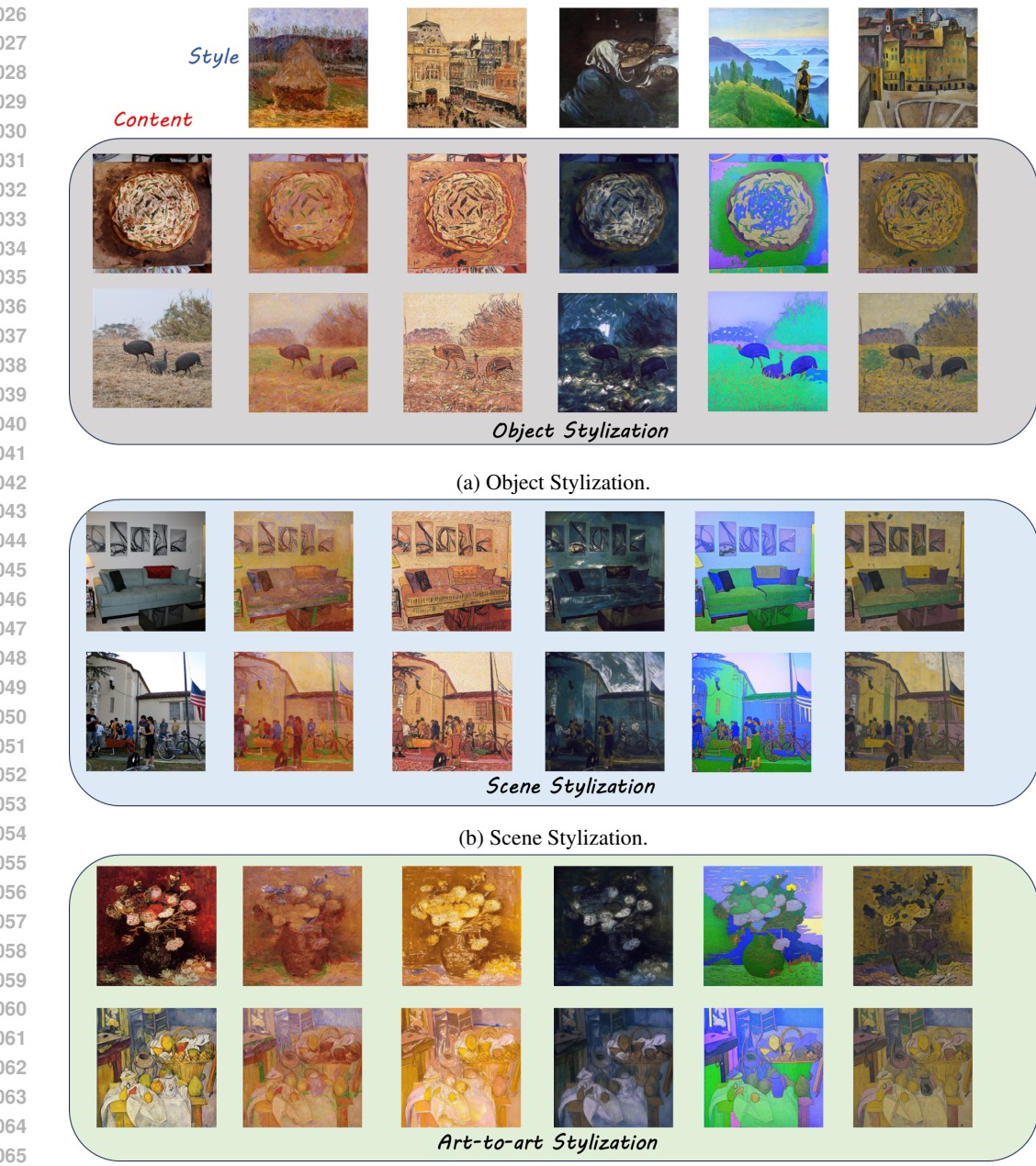

(a) Object Stylization.

(b) Scene Stylization.

(c) Art-to-art Stylization.

Figure 10: The style transfer results of object stylization, scene stylization and art-to-art transfer.

particularly valuable for artists and content creators who wish to experiment with different artistic styles while ensuring the subject remains recognizable.

## D.7 EFFECT OF INFERENCE STEP

We evaluate the effect of inference steps on the performance of our StyleBrush. Figure 13 shows a comparison between using 20 steps versus our standard 50 steps. The 20-step version shows only a modest decrease in style fidelity and identity preservation, while content similarity remains largely stable. This demonstrates that our method offers flexible quality-speed trade-offs suitable for different

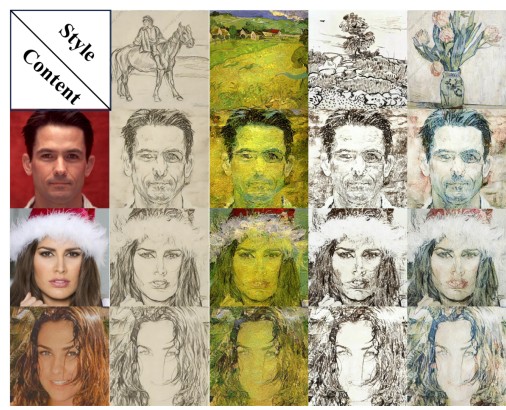 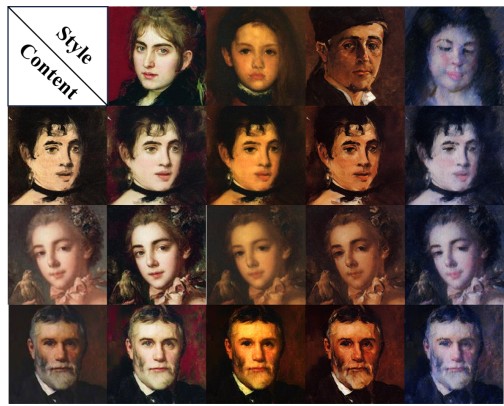

Figure 11: Style transfer results using non-facial artistic references as style images.

Figure 12: Style transfer results between different artistic portrait images.

application scenarios - users can choose 20 steps for near-real-time applications or 50 steps for highest quality results.

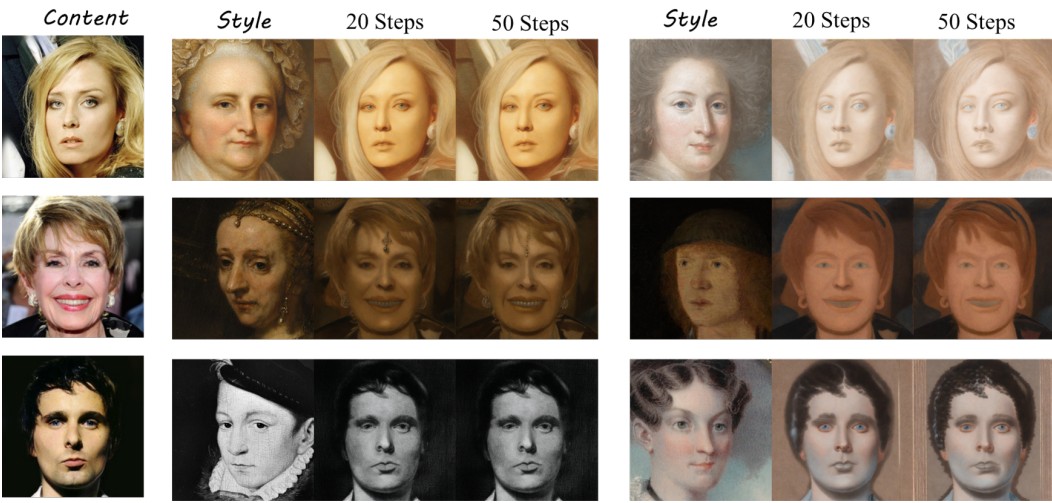

Figure 13: The style transfer results with using MetFace dataset as style reference.

## D.8 LINEAR INTERPOLATION VISUALIZATION

The pure linear interpolation of hidden states of content image, it can be seen that the interpolation is not smooth and the style is not transferred well, and the identity is not preserved well, the blur and sharp artifacts are visible.

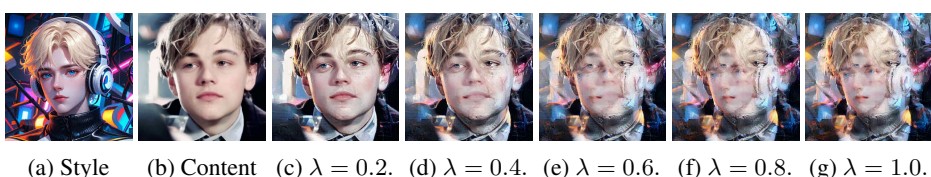

(a) Style  (b) Content  (c) $\lambda = 0.2$.  (d) $\lambda = 0.4$.  (e) $\lambda = 0.6$.  (f) $\lambda = 0.8$.  (g) $\lambda = 1.0$.

Figure 14: Linear baseline interpolation.

# E   ALGORITHM

These algorithms detail our two-stage approach for geodesic-guided style transfer. Algorithm 1 presents the geodesic-guided style interpolation process, which finds an optimal path between content and style representations in the generative model's latent space. By iteratively computing Jacobian-vector products and following the geodesic path, it achieves smooth interpolation while preserving semantic structure.

Algorithm 2 describes our style injection control mechanism, which operates in the attention space of the diffusion model. It optimizes query features through geodesic-based interpolation between content and style queries, enabling fine-grained control over style transfer while maintaining content integrity.

Algorithm 3 integrates both components into a complete style transfer pipeline. The first stage uses geodesic guidance to find an optimal interpolation point between content and style images, while the second stage performs controlled style injection through attention feature optimization. This unified approach enables both global style transfer and local feature control, resulting in more coherent and controllable stylization.

---

**Algorithm 1** Geodesic-Guided Style Interpolation

---

**Require:** Content image $x_c$, style image $x_s$, interpolation ratio $\delta$, learning rate $\lambda$
**Ensure:** Interpolated point $x_m$ on geodesic path
  1: Initialize $x_m = \delta x_c + (1 - \delta)x_s$
  2: **for** $i = 1$ to $N_{iter}$ **do**
  3:   Compute hidden state $h_m = g_d(x_m)$
  4:   Initialize random vector $v$
  5:   Compute $J_g v = \frac{\partial}{\partial a} g_u(h_m + av)|_{a=0}$
  6:   Compute $J_g^\top J_g v = \frac{\partial}{\partial h}(g_u(h_m), J_g v)$
  7:   Normalize $v \leftarrow J_g^\top J_g v / ||J_g^\top J_g v||$
  8:   Compute modified gradient $\eta_m = -\alpha v(g(x_s) - 2g(x_m) + g(x_c))$
  9:   Update $x_m \leftarrow x_m - \lambda \eta_m$
 10: **end for**
 11: Return $x_m$

---

---

**Algorithm 2** Geodesic-based Style Injection Control

---

**Require:** Content query $Q_c$, style query $Q_s$, interpolation parameter $\beta$, attention module $f$
**Ensure:** Modified query features $Q_m$
  1: Initialize mixed query: $Q_m \leftarrow \beta Q_c + (1 - \beta)Q_s$
  2: Set scaling factor $k \leftarrow \frac{1}{\beta(1-\beta)}$
  3: **for** $i = 1$ to $N1_{iter}$ **do**
  4:   $v \leftarrow$ random unit vector
  5:   Compute Jacobian-vector product:
  6:   $J_f v \leftarrow \frac{\partial}{\partial a} f(Q_m + av)|_{a=0}$
  7:   $J_f^\top J_f v \leftarrow \frac{\partial}{\partial Q_m}(f(Q_m), J_f v)$
  8:   Compute gradient:
  9:   $\nabla_{Q_m} E \leftarrow -k J_f^\top (f(Q_c) - 2f(Q_m) + f(Q_s))$
 10:   Update mixed query:
 11:   $Q_m \leftarrow Q_m - \alpha \nabla_{Q_m} E$
 12: **end for**
 13: Return $Q_m$

---

---

**Algorithm 3** Geodesic-Guided Style Transfer

---

**Require:** Content image $x_c$, style image $x_s$, content-style ratio $\delta$, query interpolation ratio $\beta$
**Ensure:** Stylized image
1: // Stage 1: Geodesic-guided style interpolation
2: Initialize $x_m = \delta x_c + (1 - \delta)x_s$
3: **for** $i = 1$ to $N_{iter}$ **do**
4:     Compute hidden state $h_m = g_d(x_m)$
5:     Initialize random vector $v$
6:     Compute $J_g v = \frac{\partial}{\partial a} g_u(h_m + av)|_{a=0}$
7:     Compute $J_g^\top J_g v = \frac{\partial}{\partial h}(g_u(h_m), J_g v)$
8:     Normalize $v \leftarrow J_g^\top J_g v / ||J_g^\top J_g v||$
9:     Update $x_m \leftarrow x_m - \lambda \nabla_{x_m} E$
10: **end for**
11: // Stage 2: Style injection control
12: Extract queries $Q_c, Q_s$ from content and style images
13: Initialize $Q_m = \beta Q_c + (1 - \beta)Q_s$
14: **for** $i = 1$ to $N_{iter}$ **do**
15:     Initialize random vector $v$
16:     Compute $J_f v = \frac{\partial}{\partial a} f(Q_m + av)|_{a=0}$
17:     Compute $J_f^\top J_f v = \frac{\partial}{\partial Q_m}(f(Q_m), J_f v)$
18:     Normalize $v \leftarrow J_f^\top J_f v / ||J_f^\top J_f v||$
19:     Update $Q_m \leftarrow Q_m - \alpha \nabla_{Q_m} E$
20: **end for**
21: // Final stylization
22: Replace query features with optimized $Q_m$ in attention layers
23: Stylized image using modified attention features

---

# F VISUALIZATION

## F.1 MORE VISUALIZATION RESULTS

In this section, we present additional visualization results to demonstrate the effectiveness and versatility of our method. Figure 15 shows additional style transfer examples with StyleBrush, Figure 9 and Figure 8 shows additional style transfer examples with StyleBrush *, illustrating how our approach successfully transfers artistic characteristics while preserving facial identity and semantic structure. The results showcase smooth transitions between content and style domains, with consistent quality across diverse artistic styles and facial features. These visualizations complement the main experimental results by providing a more comprehensive view of our method's capabilities in handling various artistic references and content images.

## F.2 MORE COMPARISON RESULTS

We provide more comparisons with state-of-the-art methods to demonstrate the superiority of our method in Figure 16.

## F.3 ZOOM-IN VISUALIZATION

To provide a detailed view of how our method handles fine-grained style transfer, we present zoom-in visualizations that highlight specific regions of interest. Figure 17 shows magnified portions of our stylization results, focusing on facial features, textures, and artistic elements. These detailed views demonstrate how our method successfully preserves important facial characteristics while incorporating artistic style elements at different scales.

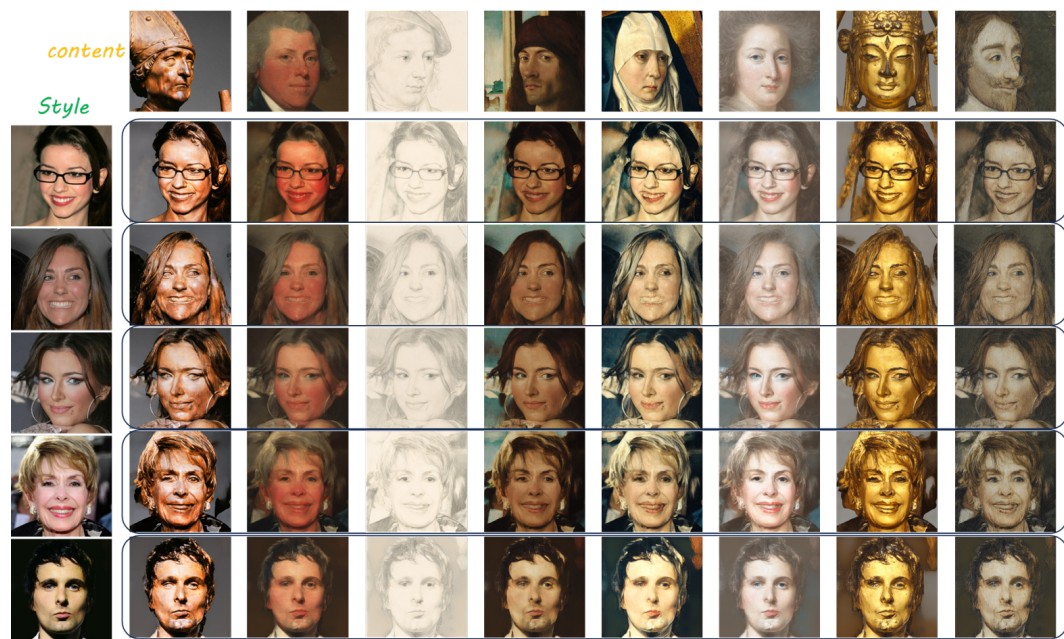

Figure 15: More visualization results showing style transfer examples with different content and style image pairs.

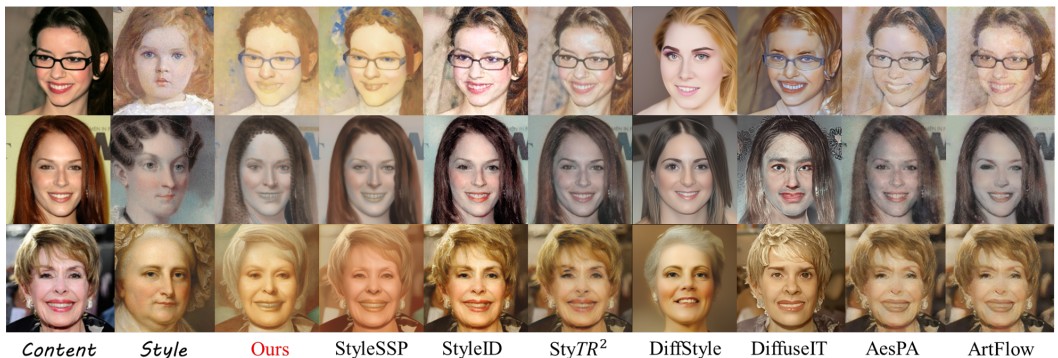

Figure 16: Visual comparison of style transfer results.

## F.4 MORE VISUALIZATION OF ARTISTIC TO ARTISTIC STYLE TRANSFER

We provide additional visualization results of artistic-to-artistic style transfer to demonstrate our method's capability in handling diverse artistic styles. As shown in Figure 18, our method successfully transfers styles between different artistic portraits while preserving the key characteristics of both content and style images.

## F.5 MORE VISUALIZATION OF NON-FACE ARTISTIC TO REALISTIC STYLE TRANSFER

We provide additional visualization results of style transfer from non-facial artistic references to realistic face images. As shown in Figure 19, our method successfully extracts and applies artistic elements from diverse non-portrait artworks while preserving facial identity and human features.

## F.6 STYLE TRANSFER RESULTS USING DIFFERENT STABLE DIFFUSION MODEL VERSIONS

To evaluate the generalization capability and robustness of our proposed method, we conduct comprehensive experiments using multiple versions of Stable Diffusion models, specifically SD v1.5, v2.0,

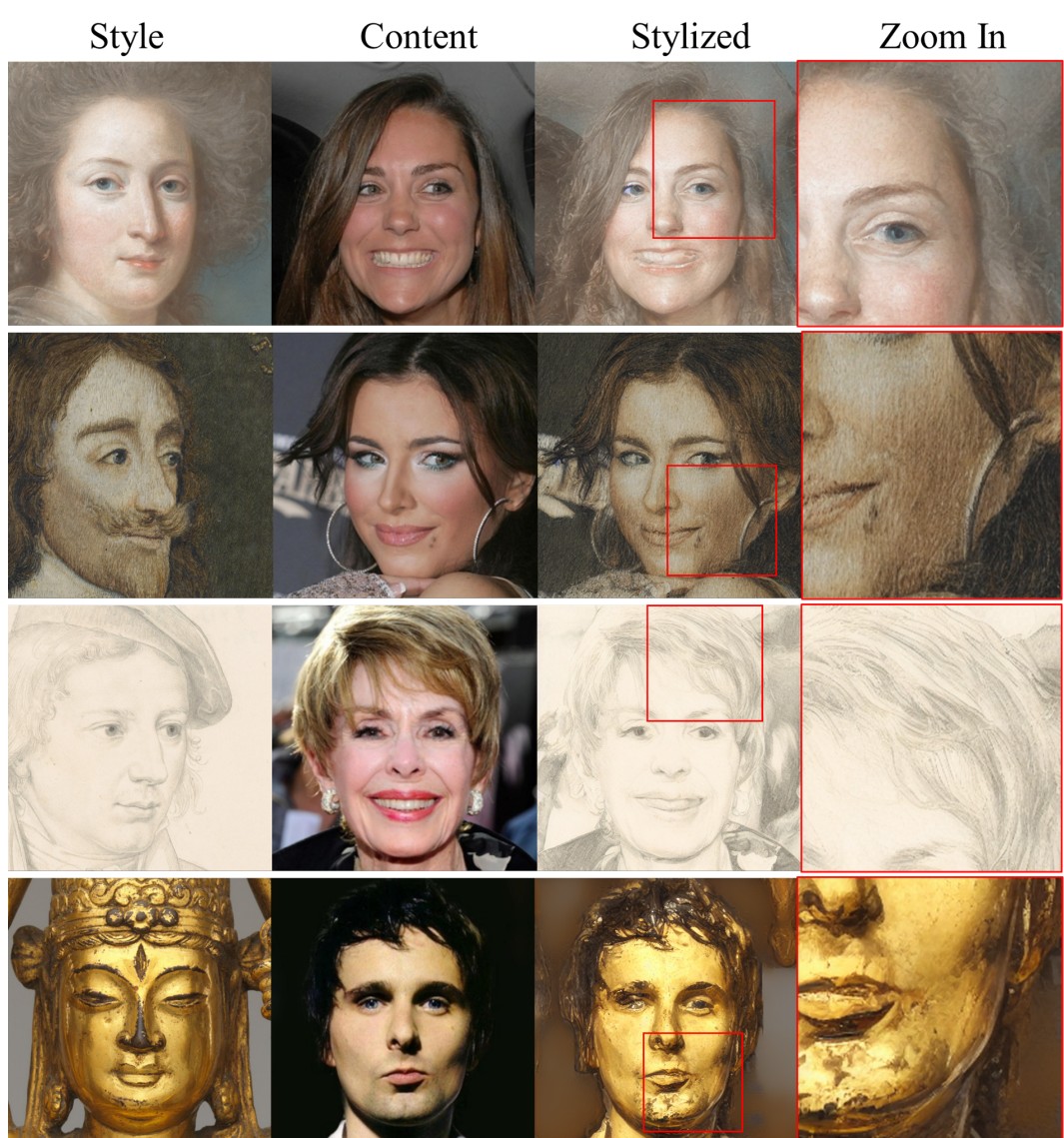

Figure 17: Zoom-in visualization comparing original and stylized image regions. The detailed views highlight our method's ability to preserve facial features while incorporating artistic elements.

and v2.1. Figure 21 and Figure 20 present a comparative analysis of style transfer results across these model variants. Our empirical findings demonstrate that the method exhibits consistent performance characteristics across different model architectures, maintaining robust facial identity preservation, semantic content fidelity, and effective artistic style element extraction and application. The consistent performance across model versions validates the architecture-agnostic nature of our approach, showing superior results compared to baseline methods, particularly StyleID. This characteristic is particularly valuable as it ensures reliable style transfer capabilities regardless of the underlying diffusion model implementation, contributing to the method's practical applicability and deployment flexibility.

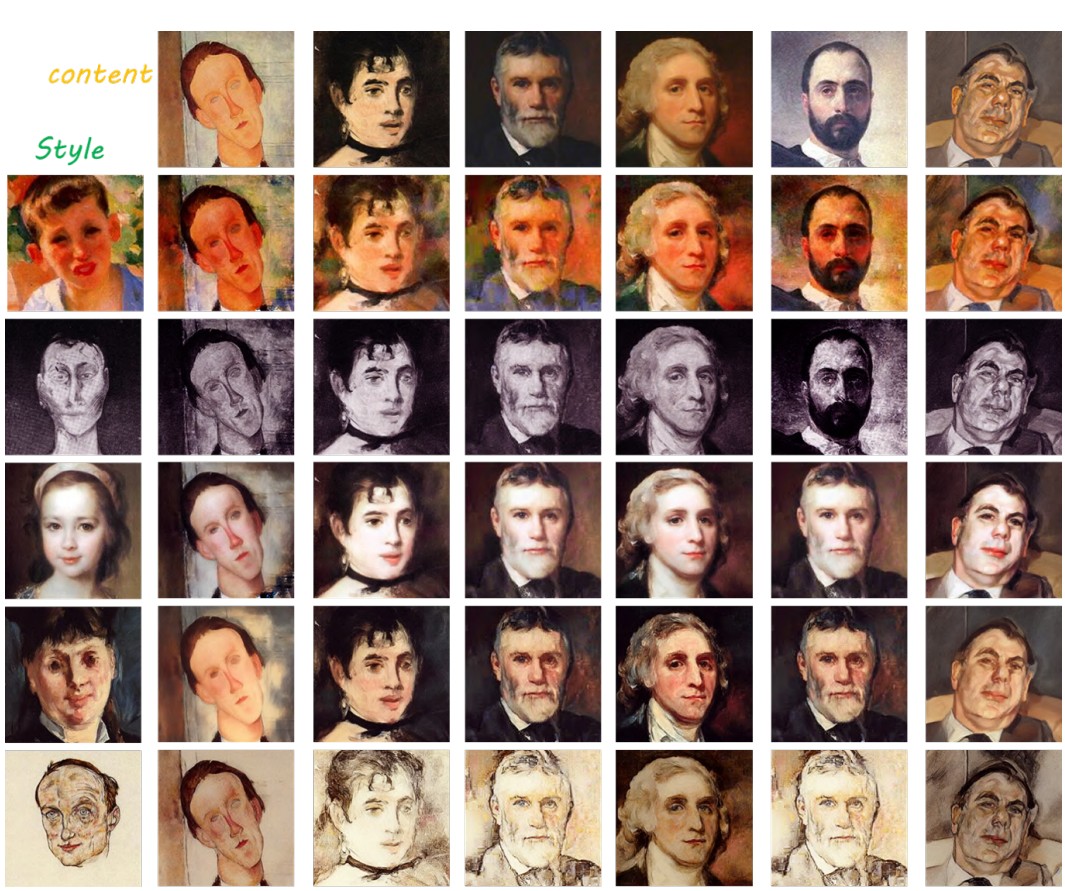

Figure 18: Additional artistic-to-artistic style transfer results. Each row shows a different example with content image, style reference, and our stylization result.

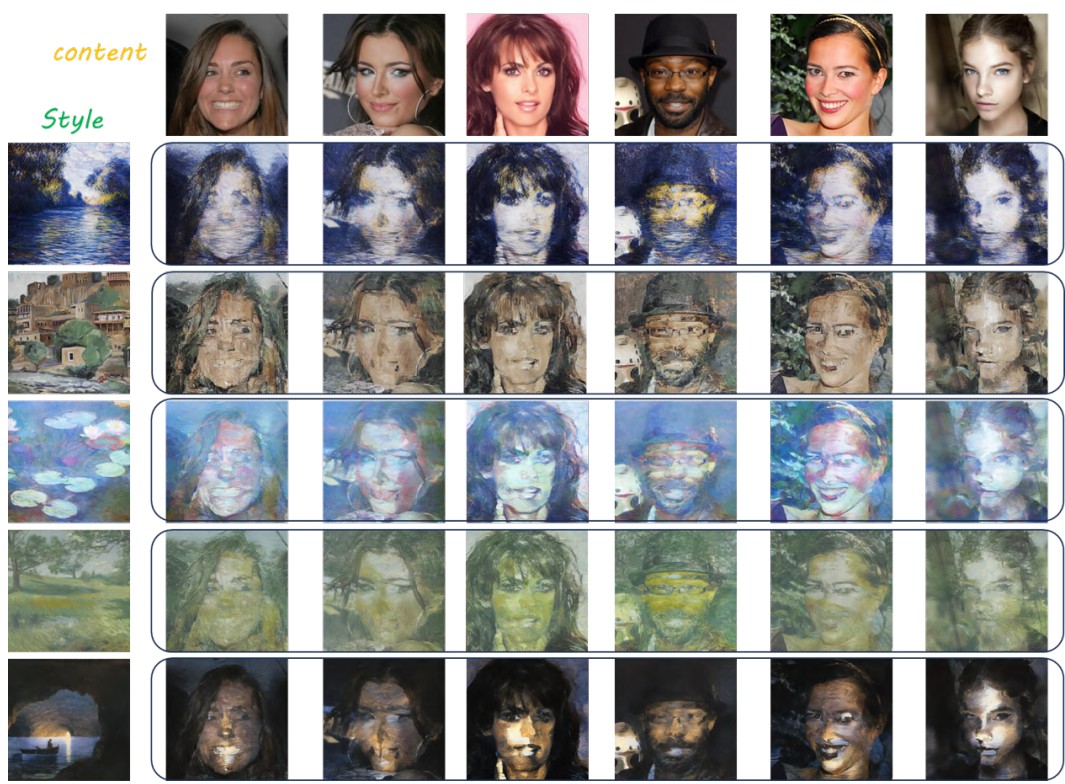

Figure 19: Additional style transfer results using non-facial artistic references. Each row shows a different example with content image, non-facial style reference, and our stylization result.

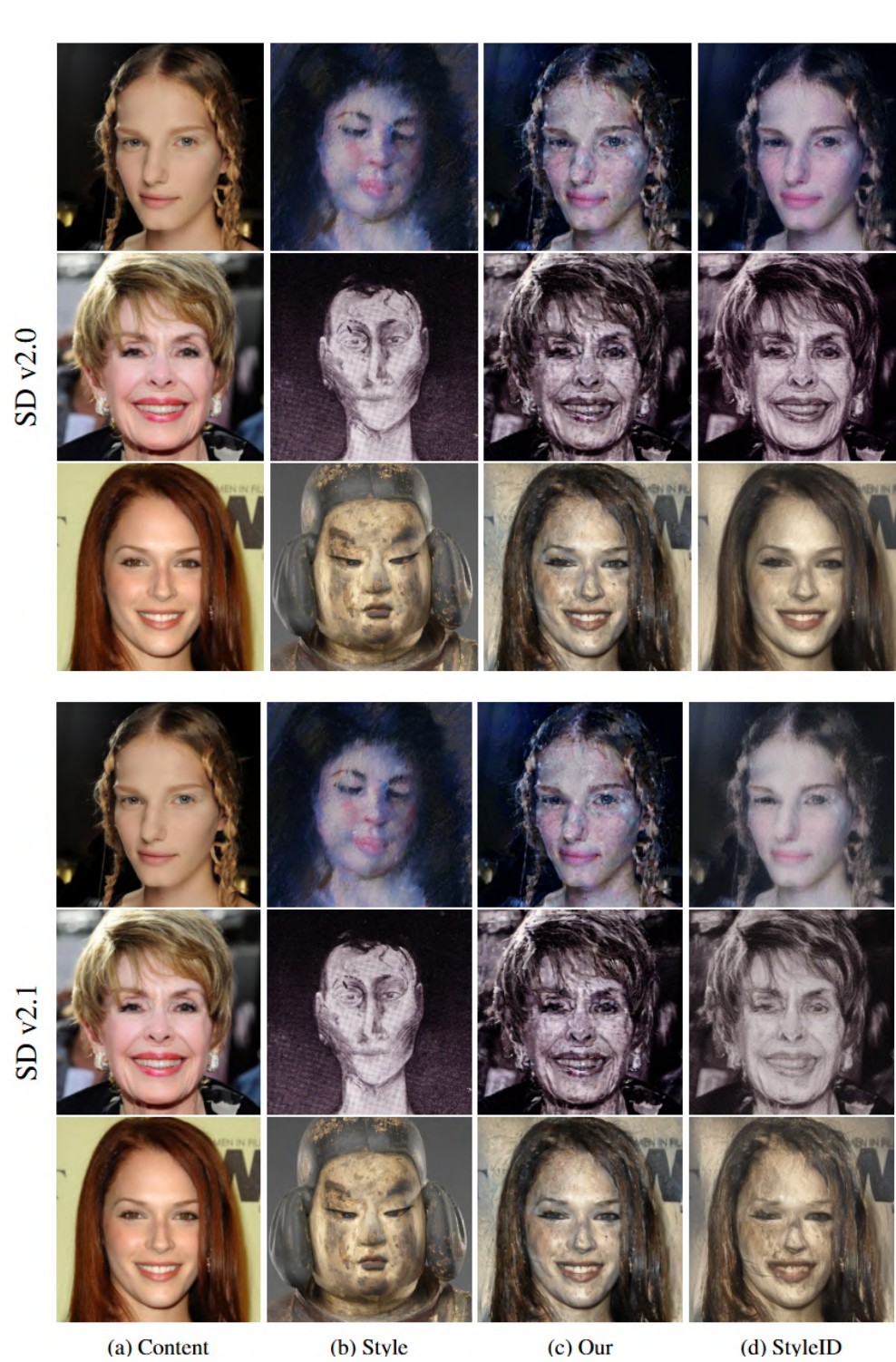

(a) Content      (b) Style      (c) Our      (d) StyleID

Figure 20: Style transfer results using different versions of Stable Diffusion models.

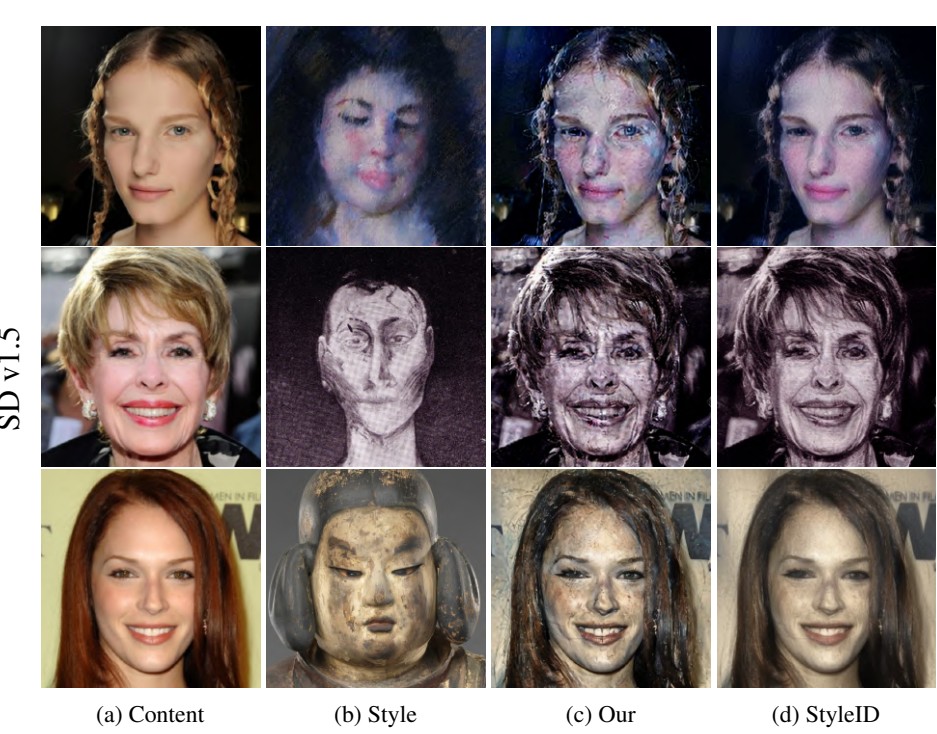

SD v1.5

(a) Content    (b) Style    (c) Our    (d) StyleID

Figure 21: Style transfer results using different versions of Stable Diffusion models.

