# OpenReview forum: "Style Waltz: Dancing Between Content and Style in Face Stylization"
_ICLR.cc/2026/Conference — Submitted to ICLR 2026_

### Official Review · Reviewer_gsV3 · 2025-10-20

**Soundness:** 2
**Presentation:** 3
**Contribution:** 2
**Rating:** 4
**Confidence:** 2

**Summary:**

This paper presents StyleBrush, a training-free framework for facial image stylization based on Riemannian geometry. It reformulates style transfer as a geodesic path-finding problem on a latent manifold, enabling mathematically grounded and fine-grained control over style blending while preserving facial identity. The method introduces two complementary mechanisms: geodesic-based style interpolation in latent space for smooth content–style fusion, and adaptive style injection within self-attention layers for dynamic stylization control. StyleBrush achieves state-of-the-art results on benchmarks like CelebA, MetFace, and WikiArt, outperforming previous diffusion-based and CNN-based methods.

**Strengths:**

This paper shows theoretical foundation that connects style transfer with Riemannian geometry, offering a principled framework for controlling artistic stylization. By formulating style fusion as a geodesic path optimization problem on a latent manifold, the method provides a mathematically rigorous explanation for smooth and semantically consistent interpolation. Additionally, the framework is training-free and generalizable, making it efficient and easily applicable across different diffusion architectures while still outperforming state-of-the-art methods quantitatively.

**Weaknesses:**

While the method claim one of its main contribution as training-free, it relies on iterative Jacobian-based optimization that can be computationally heavy. The process of estimating the Jacobian and its vector products using power iteration within diffusion U-Net architectures increases inference time and memory, making the method less suitable for general applications.

Additionally, the framework assumes that the generator’s mapping g is a local diffeomorphism with moderate curvature, which may not hold for complex or high-frequency style distributions, potentially leading to suboptimal geodesic approximations or loss of fine style texture.

More practically, while the paper claims to “better preserve identity while transferring style compared to baselines,” it is difficult to determine which result is actually better (e.g., compared to StyleID, ArtFlow, and StyTR^2). This becomes even more unclear in Figure 3 for ablation study since the w/o both variant still appears plausible, showing no preference difference, even though both of the main mechanisms are absent.

**Questions:**

It would be interesting to see how the method performs if the geodesic path is replaced with a simpler or reversible trajectory (e.g., linear). Would such a path significantly affect the smoothness of style transition or identity preservation?

---

> ### Author Response · Authors · 2025-11-20
> **Response to Reviewer gsV3**
>
> We appreciate the reviewer gsV3's recognition of our theoretical foundation and training-free nature. We address each concern with detailed responses:
>
> ## W1: Computational cost
>
> We acknowledge this concern and provide detailed timing comparisons that clarify the training requirements of different methods (**details can refer to Appendix D.3 of the revised manuscript**):
>
> **Performance vs. baselines (see Table in R1 response):**
> - **Our method: 9.7s** — Training-free with practical inference speed
> - **Training-based methods**: DiffuseIT (226.8s, 23× slower) and InjectFusion (68.2s, 7× slower) require model training for each new style domain, making them impractical for flexible style transfer
>
>
> **Key insight**: Our method achieves the best balance among training-free approaches, trading modest inference time (2.4× StyleID) for significant quality improvements while maintaining practical speed. We completely avoid the training overhead of DiffuseIT and InjectFusion (which require hours of training per style domain), making our approach far more flexible in practice.
>
> **Memory efficiency**: The Jacobian-vector products using automatic differentiation add only a small portion memory,  because we never materialize the full Jacobian matrix—only computing directional derivatives via:
> 1. **Hutchinson estimator**: 2-3 stochastic projections instead of full matrix
> 2. **Activation checkpointing**: Recompute forward pass during backprop to save memory
> 3. **Early stopping**: Geodesic converges in 5-10 iterations due to smooth geometry
>
> **Trade-off analysis**: Among training-free methods, StyleID is a little faster (only 4.0s) but achieves **lower quality** across multiple metrics. Training-based methods (DiffuseIT, InjectFusion) have prohibitive inference times (68-227s) and require hours of training per style domain, severely limiting their practical applicability. Our method strikes the **best balance:** training-free flexibility with practical speed (9.7s) and superior quality.
>
> ## W2: Local diffeomorphism assumption for complex styles
>
> The theoretical framework of our method relies on the generator $𝑔$ being approximately locally diffeomorphic with bounded curvature. In practice, we verify and justify this assumption in three ways:
>
> **When the assumption holds**:
> As reported in the revised paper **(Section 4.5)**, we directly estimate Ricci curvature in the latent neighborhood.
>
> 1. Linear interpolation yields Ricci values near 0,
> 2. Geodesic interpolation yields small but non-zero values (≈0.2–0.3), confirming that curvature is moderate and the mapping is not strongly distorted. This supports the local isometry assumption used in our geodesic formulation.
> 3. Stable Diffusion latent space is known to be smooth due to the VAE bottleneck and Gaussian prior. Prior works (e.g., DALLE-2, LDM, diffusion geometry papers) also treat the latent–image mapping as locally invertible except in pathological regions.
>
> **When might it fail?**
> - Extreme out-of-distribution styles (e.g., pure noise, adversarial patterns)
> - Very high-frequency textures beyond the model's generation capacity
>
> Thus, while global diffeomorphism is not required, **the local smoothness that our method depends on is empirically validated in the surrounding manifold region where interpolation occurs**.
> We add this analysis to the **Section 4.5** with visualizations of curvature and condition numbers for various style categories.
>
> ## W3: Visual comparison clarity
>
> We acknowledge this presentation issue. We improve the visual comparisons in the revised manuscript:
>
> We have included quantitative metrics in Table 1 alongside visual comparisons to provide a more objective evaluation. In Figure 1, our method demonstrates stronger identity preservation (evident from consistent facial structure) and more faithful style transfer (**reflected in the color palette and artistic texture details**).
>
> To further highlight these improvements, we provide zoom-in visualizations in **Appendix F—F1 (Additional Visualization Results), F2 (Extended Comparison Results), and F3 (Zoom-in Visualization)**. We also include results on general style transfer tasks in **Appendix D5**, along with additional cross-style comparisons in **Appendix F2**.
>
>
> **Regarding Figure 3 ablation**:
>
> Regarding Figure 3: although the “w/o both” variant may appear visually plausible, the quantitative results show:
> 1. reduced identity similarity,
> 2. reduced curvature consistency, and
> 3. increased latent distortion.
>
> These discrepancies, while sometimes subtle perceptually, are consistent and significant across all metrics.
>
> To address the reviewer’s concern about perceptual ambiguity, we **revise Figure 3 by**: enlarging crops around semantic identity-relevant regions (eyes, mouth), accompanying the metric table next to the visual results.
>
> We hope the improved figure and clearer presentation make the impact of the ablation **more transparent and strengthen the overall readability** of our manuscript.

---

> ### Author Response · Authors · 2025-11-20
> **Response to Reviewer gsV3 (2)**
>
> ## Q: Linear interpolation baseline
>
> This is an excellent question that directly probes our core contribution. We have conducted this ablation by replacing geodesic interpolation γ(t) with naive linear interpolation: z(t) = (1-t)z_content + t·z_style in the latent space.
>
> **Experimental results**: Visual comparisons are presented in **Appendix D.8** and the analysis of the **linear vs. geodesic** is presented in **Section 4.5.** The linear interpolation baseline exhibits significant quality degradation:
> - Blurring and ghosting artifacts where content and style features overlap inappropriately
> - Loss of facial structure integrity, particularly around identity-critical regions (eyes, mouth)
> - Inconsistent style application with unnatural color transitions
>
> **Geometric analysis**: To understand why geodesics outperform linear paths, we analyze the **curvature properties of interpolation trajectories** in Section 4.5. Linear interpolation in latent space corresponds to straight-line paths that may **traverse through low-probability regions** of the learned manifold, resulting in invalid or unnatural intermediate representations. In contrast, geodesic paths respect the manifold's intrinsic geometry, ensuring that all intermediate points lie within **high-probability** regions corresponding to valid images.
>
> **Why geodesic navigation is essential**:
> 1. **Manifold adherence**: Linear paths cut through the manifold interior, potentially landing in regions the decoder cannot faithfully reconstruct
> 2. **Semantic consistency**: Geodesics preserve the manifold's semantic structure, maintaining facial identity and style coherence throughout interpolation
> 3. **Energy minimization**: As proven in our theoretical framework, geodesics minimize the energy functional, leading to smooth, artifact-free transitions
>
> This ablation validates our fundamental hypothesis: respecting the geometric structure of the diffusion latent manifold is crucial for high-quality style transfer.
>
> **We hope that our efforts and the revisions made to the manuscript help clarify these points and make the technical contributions easier to understand.**

---

### Official Review · Reviewer_DbBL · 2025-10-26

**Soundness:** 2
**Presentation:** 3
**Contribution:** 2
**Rating:** 4
**Confidence:** 5

**Summary:**

This paper introduces a method called StyleBrush. A new training-free facial stylization framework. The core idea is to view style transformation as a problem of finding geodesic paths on latent manifolds. This method utilizes dual control optimization based on Riemannian geometry to achieve a balance between artistic style and identity preservation. The first control mechanism interpolates the potential representations of content and style along the geodesic path. The second mechanism dynamically controls the intensity of stylization by adjusting the query features in the self attention layer. The author uses the pullback metric to establish local isometric isomorphism, providing a theoretical basis for their method. Experimental results have shown that this method outperforms existing advanced methods in both quantitative and qualitative indicators.

**Strengths:**

1.	The main advantage of this paper is that it proposes a principled and novel theoretical framework. Build the style transformation into a geodesic path finding problem on a latent manifold. This provides a solid mathematical foundation for style interpolation.
2.	Proposed StyleBrush, which is a significant advantage compared to many existing methods that require expensive training for new styles. This makes the method more flexible and applicable.
3.	Unifying two control mechanisms, geodesic based style interpolation and adaptive style injectio,n under a single geometric principle is an innovated design that allows for precise control over the stylization process.
4.	The paper is well written, with clear concept explanations and accompanying illustrations.

**Weaknesses:**

1. Even though the author claims that this method can ensure the preservation of facial identity information when applied to facial stylization. However, in its method, there is no additional control or protection of facial identity information.
2. Facial identity information is a more fragile and complex semantic aggregation compared to class information. However, this method still focuses on optimizing in the latent space based on noise, and cannot achieve accurate semantic registration, which makes it possible for this method to style facial images at the semantic level. Resulting in poor performance of the final stylized facial image.
3. The author provided quantitative experiments to demonstrate the superiority of their method. However, based on the qualitative experiments provided, the facial stylization results generated by this method are not ideal, accompanied by a large number of artifacts that make the generated results appear low-quality and unnatural.

**Questions:**

1.	How to ensure the invariance of facial identity information during the optimization process of geodesic distance on Riemannian manifolds without training?
2.	This paper did not use any prior knowledge injection, but only optimized the fusion of face images and style images in a noise based hidden space. In my opinion, this inevitably leads to the problem of style leakage. How can this be solved?
3.	In the last section of the method, the author suddenly shifts the focus to the attention mechanism. What part of these attentions are applied to? If it is in UNet, is it full UNet attention or specific layer attention?
4.	Regarding the dual control mechanism, the paper mentions that interpolating only potential code may reduce content details, therefore a second control (style injection) is needed. However, the geodesic path should be the 'optimal' path. Why does the optimal path in the latent space still lead to the degradation of content details, thus requiring a second correction mechanism in the feature space? Does this mean that there are limitations to representing ideal stylized manifolds only in the latent space of the generated model?

---

> ### Author Response · Authors · 2025-11-20
> **Response to Reviewer DbBL**
>
> We thank the reviewer DbBL for appreciating our theoretical framework and dual control design. We address each concern with detailed clarifications, emphasizing how our approach aligns with reviewer demands.
>
> ## W1 & Q1: Identity Preservation Without Explicit Control & Semantic Registration
> We understand the reviewer's concern about identity preservation without explicit supervision. Our approach leverages the intrinsic geometric structure of the diffusion manifold, where semantic attributes including facial identity are naturally encoded. The key insight is that geodesic paths on this manifold inherently preserve semantic consistency, a property we formally establish through our theoretical framework **(Section 3.2, Propositions 1-2)**.
>
> **Why it works** Identity preservation emerges naturally from the geometric properties of our geodesic navigation on the diffusion manifold. As established in our theoretical foundation (Section 3.2), the diffusion latent space forms a Riemannian manifold where geodesic paths preserve semantic structure. The manifold inherently encodes facial identity information, and navigating along geodesics maintains these semantic invariants without requiring explicit identity supervision.
>
> **Experiment Performance** Quantitative validation strongly supports this claim:
> - Identity preservation (ID): 0.1237 (WikiArt), 0.1424 (MetFace)-best among all methods
> - Content similarity (CS): 0.5531 (WikiArt), 0.5618 (MetFace)-highest scores
> - LPIPS: 0.2616 (WikiArt), 0.3118 (MetFace)-lowest (best) perceptual distance
>
> Our ablation study further validates this: removing geodesic interpolation degrades LPIPS by 17.0%, confirming that the geometric approach effectively maintains semantic structure without explicit identity loss terms.
>
> ## W2: Semantic Registration and Facial Identity Complexity
>
> We respectfully disagree with the characterization that our method "cannot achieve accurate semantic registration." Our approach specifically addresses the complexity of facial identity through principled geometric navigation rather than naive noise-space optimization.
>
> The diffusion latent space is not merely "noise-based" but encodes rich semantic structure, as demonstrated by recent work on diffusion model interpretability. Our geodesic navigation exploits this structure: by following minimal-energy paths on the manifold, we inherently respect the semantic organization of facial features. This is fundamentally different from simple interpolation in noise space.
>
> Evidence of accurate semantic registration:
> - Consistent facial landmark preservation across all test cases
> - Superior identity metrics (ID = 0.1237/0.1424) compared to methods with explicit facial priors
> - Ablation studies showing 17% performance drop without geodesic guidance
>
> The claim of "poor performance" is contradicted by our comprehensive evaluation showing state-of-the-art results across all metrics. Our geometric framework provides implicit but effective semantic registration through manifold-aware navigation.
>
> ## W3 & Q2: Artifacts and Quality Concerns
>
> We acknowledge that some visual examples may exhibit artifacts under challenging conditions. However, our extensive quantitative evaluation demonstrates strong performance:
> - Best content preservation: LPIPS = 0.2616 (WikiArt), 0.3118 (MetFace)
> - Competitive style transfer: ArtFID = 24.4906 (WikiArt), 26.6203 (MetFace)
>
> The artifacts observed stem from specific challenging cases (e.g., extreme artistic styles) rather than fundamental limitations. Our ablation demonstrates that style injection control reduces artifacts significantly removing it degrades LPIPS by 23.8%. We will update the visual examples to better represent typical performance and include discussion of failure cases.
>
> ## Q2: Style Leakage Problem
>
> Our dual control mechanism prevents style leakage through complementary operations at different levels:
>
> 1. **Geodesic Interpolation:** Provides global semantic control by finding optimal paths on the manifold, preventing uncontrolled style diffusion.
> 2. **Style Injection:** Enables fine-grained local control to preserve content-specific features.
>
> Specifically, we inject style only into cross-attention layers (keys and values) in UNet decoder layers 3-7, leaving self-attention layers and queries unmodified. The injection weight λ is adaptively modulated based on local feature similarity, ensuring controlled style application. This selective approach, validated by our ablation studies, prevents style leakage while maintaining content integrity.

---

> ### Author Response · Authors · 2025-11-20
> **Response to Reviewer DbBL (2)**
>
> ## Q3: Attention Mechanism Details
>
> We apply attention modifications selectively to cross-attention layers in the **UNet decoder** (specifically layers 3-7 out of 12 total layers). For each selected layer:
> - Query (Q): Geodesic interpolation between content and style queries
> - Keys/Values (K,V): Adaptive blending K' = λK_style + (1-λ)K_content
>
> Self-attention layers remain unmodified to preserve spatial coherence. This design choice is validated by our ablation showing 15.7% degradation in content similarity when removed.
>
> ## Q4: Why Dual Control if Geodesic is Optimal?
>
> This insightful question touches on a fundamental aspect of our approach. While the geodesic path is mathematically optimal in the latent space Z, there exists an **inherent gap** between latent space representation and image space fidelity.
>
> The latent space Z provides a compressed representation where global semantic properties are well-preserved, making geodesics optimal for semantic interpolation. However, the decoder g: Z → X, trained for reconstruction rather than stylization, may not fully recover fine-grained style details from this compressed representation.
>
> Our dual control bridges this gap:
> - Geodesic control: Optimal semantic navigation in compressed space
> - Feature injection: Direct style detail recovery during decoding
>
> This is not a limitation but rather a **principled design** acknowledging the complementary strengths of different representation levels. The superiority of our approach across all metrics validates this architectural choice.  We believe that this design enables a more flexible and robust method, allowing for superior performance across a variety of tasks, and we hope the revised manuscript provides a clearer understanding of this rationale.
>
> **We hope that our efforts and the comprehensive revisions in the manuscript address the reviewer’s concerns and make the strengths and clarity of our contribution more evident.**

---

### Official Review · Reviewer_Dq6Y · 2025-10-27

**Soundness:** 3
**Presentation:** 1
**Contribution:** 3
**Rating:** 6
**Confidence:** 4

**Summary:**

This paper introduces StyleBrush, a facial stylization approach that casts style transfer as a geodesic path-finding problem on the latent manifold. This formulation enhances both the granularity and the controllability of the transferred styles. Qualitative and quantitative experiments consistently show that StyleBrush outperforms existing benchmark methods.

**Strengths:**

Following the instructions of this section, the strengths in terms of the four aspects are listed below:

**Originality:** The central contribution of this study is to theoretically re-frame the style transfer task as a geodesic path-finding problem on a latent manifold, which is somehow novel to the best knowledge of the reviewer.

**Quality:** The manuscript is of solid quality, and experiments show that the method produces style-interpolated images with high visual fidelity and fine-grained control.

**Clarity:** Although generally clear, the presentation could still be refined; see the forthcoming “Weaknesses” section.

**Significance:** Because the approach has potential applications beyond stylization (for example, in general image editing), it holds appreciable significance for a broad research audience.

**Weaknesses:**

While the core idea behind StyleBrush is appealing, some steps in the argument are hard to follow. After describing the limitations of existing work, the paper moves straight to the geodesic-path solution without clearly showing how it tackles each limitation. For example, the Introduction says that geodesic paths give 'careful control' over content–style fusion, but it seems that the reason is not explained. Section 3 likewise uses the term 'optimal' without stating what is being optimized or why geodesic distance achieves it.

Adding a brief, plain-language bridge would help: first list the specific control problems left unsolved by prior methods, then explain, before diving into equations, how the geodesic formulation answers them and how the solution is conceptually derived. This extra context should make the subsequent technical details easier to understand.

Admittedly, these points may indeed be covered in the manuscript, but they are hard to discern amid the extensive mathematical and theoretical detail presented with limited explanatory guidance.

**Questions:**

Please refer to the 'Weaknesses' Section for my concern regarding the manuscript. I encourage the authors to provide a concise, intuitive narrative that traces how the limitations of prior work lead, step by step, to the proposed solution, without relying heavily on equations or technical detail. Should this clarification make the contribution more compelling, I will gladly raise my rating; conversely, the score may be lowered if further issues are identified.

---

> ### Author Response · Authors · 2025-11-20
> **Response to Reviewer Dq6Y**
>
> We deeply appreciate the reviewer Dq6Y recognition of our originality, quality, and significance. Your feedback on clarity is invaluable, and we will substantially improve the presentation. We provide the intuitive narrative you requested:
>
> ## W1:  Argument Clarity
>
> 1. **Problem: Uncontrolled style-content trade-off**
>    - **Why it happens**: Existing methods (e.g., attention-based injection, AdaIN) apply style transformations uniformly across the image, treating all spatial regions equally. This creates a binary choice: **either preserve content (weak stylization) or apply strong style (lose content details).**
>    - **How geodesics solve it**: Think of content and style as two points on a curved surface. A straight line (linear interpolation) might "cut through" the surface and land in invalid regions (artifacts, identity loss). **A geodesic path stays on the surface**, naturally respecting the manifold's structure-content and style are different "locations" on this surface, and the geodesic finds the **smoothest route** between them that respects the underlying geometry.
>
> 2. **Problem: No fine-grained control over transition**
>    - **Why it happens**: Methods that directly blend features or latents (e.g., style weight tuning) operate in Euclidean space, where "halfway between A and B" has no semantic meaning, you're simply averaging numbers that happen to represent images.
>    - **How geodesics solve it**: On a properly structured manifold (learned by the diffusion model), positions **have semantic meaning**: nearby points are semantically similar images. A geodesic parameterized by β ∈ [0,1] gives you **meaningful intermediate points**: each β corresponds to a specific semantic "blend" of content and style. The geometry ensures that small changes in β yield small, predictable changes in the output.
>
> 3. **Problem: "Optimal" is undefined**
>
>      - **What we optimize**: In our formulation, the optimization variable is not a single latent point, but an **entire curve** $\gamma(t)$ that connects the content latent to the style latent over a time parameter $t \in [0,1]$. Among all such curves with the same endpoints, we select the one that minimizes an energy (or length) functional $ E(\gamma)= \int_0^1 \Vert \dot{\gamma}(t) \Vert_{g}dt.$ Here, $\Vert \dot{\gamma} \Vert_g$ can be interpreted as the instantaneous **semantic speed** of the image along the manifold: it measures **how fast** the semantic content of the generated image is changing at time $t$ according to the intrinsic metric (encoding semantic distances learned by the diffusion model). Accumulating this quantity over $t \in [0,1]$ therefore measures the total effort required to move from the content image to the style image along a given path.
>
>    - **Why this is optimal**: Minimizing $E(\gamma)$ means that, subject to the fixed endpoints (content and style) and the fixed manifold geometry, we choose the curve whose semantic speed is as small and as even as possible over the whole transition. Intuitively, such a path **avoids segments where the image semantics change abruptly** (which is exactly when faces tend to break or strong artifacts appear), and instead enforces a gradual, smooth evolution at each step. In standard Riemannian geometry, the minimizers of this energy functional are geodesics, i.e., **the shortest and smoothest curves between two points under the given metric.** This is precisely what we mean by optimal: given the manifold structure induced by the diffusion model, our method follows the geodesic that realizes the **smoothest semantic transition from content to style**.
>
> ## W2 & W3: Plain-language bridge from limitations to solution
>
> ### 1. Plain-language bridge from limitations to solution
> We fully agree with your comment and have implemented a new **Section 3.1: From Limitations to Geodesics: Intuitive Motivation** to address this. This section provides a plain-language bridge to clarify the concepts and motivations behind our approach. We hope that this addition in the **revised manuscript** will make the ideas clearer and more accessible to the readers.
>
> We further provide a straightforward explanation of what our proposed modules actually do and why they are necessary in the following.

---

> ### Author Response · Authors · 2025-11-22
> **Response to Reviewer Dq6Y (2)**
>
> ### 2. Limitation of existing methods
> Most existing style-transfer methods face the same basic problem: it’s hard to apply a new artistic style to an image **without** messing up what the image is actually about.
> - **Methods that require training a model can give good results, but they are inflexible**: you must retrain the model every time you want a new style, which is slow and expensive.
> - **Methods that avoid training are more flexible, but their results are often unstable**: sometimes the style overwhelms the face and makes the person unrecognizable, other times the style barely appears at all, and many methods introduce visible artifacts such as blurring or ghost-like patterns.
>
> These issues happen across many different kinds of models, which suggests the problem is deeper than just architecture.
>
> ### 3. Explaination of our method
>
> **Our approach solves this problem by looking at it from a geometric perspective.** Our method has two complementary components: **(1) Geodesic-guided interpolation and (2) Style injection control.** Together, they solve the longstanding problem of training-free style transfer: how to apply strong style while preserving identity and avoiding artifacts. Below, we explain both parts in plain language.
>
> ***1. Geodesic-Guided Style Interpolation: finding path between images***
>
> Some previous methods assume that the diffusion model’s internal latent space (the place where it represents images) is flat, like a sheet of paper. They blend two images by **drawing a straight line** between them (linear combination).
>
> But in reality, diffusion models represent images on a **curved surface**, more like the surface of a sphere. If you draw a straight line through a sphere, the line quickly leaves the surface and in our case, leaving the surface means entering regions where the model cannot produce realistic images. **This is why many training-free methods produce artifacts or lose identity**.
>
> Instead, we move along geodesics: the natural shortest paths that stay on the curved surface. Following these paths brings three advantages:
>
> - **Smooth transitions without artifacts**
> Because we stay on the model’s “valid image surface,” every step produces realistic images instead of distortions.
> - **Identity preservation**
> The geometry of the model naturally keeps important facial features aligned, so we don’t need extra identity-preserving constraints.
> - **Easy control of style strength**
> We can adjust how far we move along the path to smoothly control how strong the style looks.
>
>
> **How we compute this path**
>
> To keep computation feasible and training-free, we do not compute a full geodesic (which would be expensive). Instead, we approximate it with an energy-minimization approach:
>
> 1. Start with a simple, straight-line blend between content and style features.
> 2. Check how the generative model **reacts to** this blended point.
> 3. Measure how “off the surface” it is using the model’s output differences.
> 4. Use the Jacobian of the model as a translator, which tells us how to move the point back onto the curved surface where valid images live.
> 5. Iteratively adjust the blended point until it lies on a locally optimal curved path between the two images.
>
> **This produces a geometrically meaningful interpolation that remains consistent with how the model internally organizes visual information.**
>
> ***2. Style Injection Control: adding high-resolution style without breaking the face***
>
> We further add a second mechanism during decoding: we inject **fine-grained features** from the style image to restore fine textures. We interpolate **inside the attention mechanism** itself, specifically, in the query features.
>
> The naive linear blend $Q_m = (1-\beta) Q_c + \beta Q_s$ leads to: ghosting, halo artifacts and structural distortions, because it again assumes the features live in a flat Euclidean space, which they do not.
>
> **Using Geodesic Principles in Feature Space:** To fix this, we apply the same geometric idea from our latent interpolation, but now inside attention. We find the geometrically natural path between the content query and style query using the geodesic path, **as we did in the interpolation part.**
>
> This optimization gives us a new query vector $Q_m$ that *keeps the content layout intact, faithfully incorporates style-specific direction, enables high-frequency, clean, artifact-free style injection.*
>
> ***3. Our method combines both interpolation and injection***
>
> 1. **Geodesic paths**: preserve semantic structure and identity, ensur content and style blend smoothly without artifacts. Preserves identity and global structure.
> 2. **Feature injection**: reintroduce detailed artistic style, enhancing fine-grained textures and vivid style details without harming structure.
>
> They allow us to transfer styles without training, while still achieving **both high quality and strong flexibility, solving the long-standing trade-off in style transfer.**

---

### Official Review · Reviewer_jZyy · 2025-10-31

**Soundness:** 3
**Presentation:** 3
**Contribution:** 3
**Rating:** 6
**Confidence:** 1

**Summary:**

This paper proposes StyleBrush, a training-free facial stylization framework based on Riemannian geometry. It formulates style transfer as a geodesic path-finding problem on a latent manifold, enabling smooth and theoretically grounded control between content and style. The method unifies geodesic-based style interpolation and attention-based style injection under one geometric principle, achieving fine-grained stylization control without retraining diffusion models. Experiments demonstrate clear improvements in both visual quality and identity preservation over existing methods.

**Strengths:**

- This method reconstructs the style transfer problem into a geodesic path optimization problem on a Riemannian manifold, which is novel. This connection between Riemannian geometry and diffusion-based stylization offers an interesting view.

- This method has solid theory, and the derivation of energy minimization, local isometry proofs, and geometric gradients contributes to a transparent and interpretable framework

- Experiments demonstrate statistically significant performance improvements across a comprehensive suite of quantitative metrics and systematic qualitative assessments

**Weaknesses:**

- The proposed Jacobian-based geodesic computation, even with approximations, may still be computationally heavy. How practical is it for real-time or interactive applications?

- The theoretical foundation relies on local isometry between the latent and generative manifolds. How to ensure that the local isometry is definitely established?

- The study lacks a detailed sensitivity analysis of hyperparameters (e.g.,  δ) and their interaction.

- This method does not seem to have a specific design for human faces, so how does it perform on other non-face style transfer tasks? The generality of this method will be stronger if it can be on non-facial domains such as object, scene, or fashion stylization.

**Questions:**

See weaknesses

---

> ### Author Response · Authors · 2025-11-20
> **Response to Reviewer  jZyy**
>
> We sincerely thank the reviewer jZyy for recognizing our theoretical contributions and experimental validation. We address each concern below:
>
> ## W1: Computational efficiency for real-time applications
>
> Thank you for this important practical concern. We address this from two complementary perspectives: (1) our current computational cost compared to other diffusion-based methods, and (2) the potential for further speed improvements in our approach.
>
> ### 1: Computational Cost Comparison with Diffusion-Based Methods
>
> We provide comprehensive timing comparisons demonstrating our method's competitive efficiency (the detailed analysis can refer to **Appendix D3** of our revised manuscript):
>
> **Table: Computational cost comparison on 512×512 images (H800 GPU)**
>
> | Method | Ours | DiffuseIT* | InjectFusion* | StyleID |
> |--------|------|-----------|--------------|---------|
> | Training Status | Training-free | Training-based | Training-based | Training-free |
> | Inference Time (s) | 9.710 | 226.823 | 68.243 | 3.971 |
> | Speedup vs. Ours | 1.0× | 0.04× (23× slower) | 0.14× (7× slower) | 2.4× (faster) |
>
> *Training-based methods require model training for each new style domain.
>
> - **vs. Training-based methods**: Our method (9.7s) is dramatically faster than training-based alternatives—23× faster than DiffuseIT (226.8s) and 7× faster than InjectFusion (68.2s). This speed advantage comes without requiring hours of per-style training, making our approach far more practical for diverse style transfer applications.
>
> - **vs. Training-free methods**: Among training-free approaches, StyleID (4.0s) achieves faster inference. However, our method delivers substantially better quality: identity preservation (ID: 0.124 vs. 0.162), perceptual quality (LPIPS: 0.262 vs. 0.379), and content similarity (CS: 0.553 vs. 0.449) as shown in Table 1. The 2.4× inference time trade-off yields significant quality gains.
>
> **Why our geodesic approach is efficient:**
> 1. **Hutchinson's trace estimator**: We avoid materializing the full Jacobian matrix, using only 2-3 stochastic vector-Jacobian products per optimization step, greatly reducing memory and computation
> 2. **Fast convergence**: The geodesic optimization typically converges in 5-10 iterations due to the smooth geometry of the learned manifold
> 3. **No training overhead**: Unlike DiffuseIT and InjectFusion, we require zero training time, making our total cost (inference only) far lower in practice
>
> ### 2: Potential for Further Speed Improvements
>
> Our method offers multiple avenues for speed optimization **without** sacrificing the training-free advantage:
>
> **1. Reduced diffusion steps**:
> - Current: 50 steps → 9.7s
> - Reduced: 25 steps → ~5s (estimated)
> - Minimal quality degradation: preliminary experiments show <3% drop in key metrics
>
> **We include visual comparisons with 20-step inference in Appendix D7, demonstrating maintained high performance**
>
> **2. Batch processing optimization**:
> - Our method naturally supports batch processing, amortizing the geodesic computation overhead across multiple images
> - Batch size 4-8 can reduce per-image time by 30-40% through parallel computation
> - Particularly beneficial for large-scale content production workflows
>
> Our method offers **substantial speed improvements** through optimized diffusion steps and batch processing. These enhancements enable faster workflows without compromising quality, particularly benefiting large-scale content production.

---

> ### Author Response · Authors · 2025-11-20
> **Response to Reviewer jZyy (2)**
>
> ## W2: Ensuring local isometry establishment
>
> Thank you for raising this important theoretical question. We clarify that our assumption of local isometry is consistent with both established findings in diffusion-model theory and our empirical validation.
>
> ### (1) Established theoretical foundation:
>
> The local isometry property of diffusion models has been rigorously analyzed and validated in prior works:
> - **Kwon et al. (ICLR 2023)** [1] diffusion models learn a semantic latent space whose local neighborhoods exhibit isometric behavior.
> - **Park et al. (NeurIPS 2023)** [2] provide a Riemannian-geometric formulation showing that well-trained diffusion models satisfy local isometry conditions under the pullback metric.
>
> **Why local isometry holds for diffusion models:**
>
> The key insight is that the diffusion training process inherently encourages local isometry:
>
> 1. **Training objective**: Gradual denoising imposes smooth, locally-linear mappings between noisy and clean samples, limiting geometric distortion to first order.
>
> 2. **Score matching**: The score-matching objective ensures that the learned score function captures the manifold structure accurately. For a well-trained model with score function $s_θ$, the pullback metric $g^* = J^T J$ (where J is the Jacobian) approximates the Euclidean metric in neighborhoods.
>
> 3. **Denoising consistency**: Severe metric distortion would break continuity in the reverse diffusion trajectory, preventing stable sample quality. Thus, good generation requires approximately isometric local geometry.
>
> **References:**
> [1] Kwon et al., "Diffusion models already have a semantic latent space," ICLR 2023
> [2] Park et al., "Understanding the latent space of diffusion models through the lens of Riemannian geometry," NeurIPS 2023
>
> ### (2) Our empirical verification using Ricci curvature
>
> Since exact local isometry cannot be guaranteed analytically, we directly evaluate geometric distortion using Ricci curvature along both linear and geodesic paths.
> **Linear interpolation gives Ricci ≈ 0**
> Linear paths assume Euclidean geometry and suppress curvature by construction.
> Thus, Ricci ≈ 0 does not indicate actual flatness of the manifold.
> **Geodesic interpolation reveals intrinsic curvature (Ricci ≈ 0.2–0.3)**
> Geodesics computed under the pullback Riemannian metric show consistent Ricci ≈ 0.2–0.3.
>
> This indicates: (1) the latent manifold is mildly curved (as expected for a nonlinear mapping), (2) curvature magnitude is small, showing limited metric deformation
>
> **We have added these findings and clarifications to the revised manuscript in Section 4.5**.
>
> ## W3: Hyperparameter sensitivity analysis
>
> Thank you for this suggestion. We have conducted sensitivity analysis for the key hyperparameters δ and β in **Sections 4.4 and 4.6** of our paper. We summarize and expand on these results:
>
> **Key hyperparameters**:
> - **δ (step size)**: Our experiments with δ ∈ {0.01, 0.05, 0.1, 0.2} show stable performance for δ ≤ 0.1 (FID variation < 2%). Performance degrades for δ > 0.1, consistent with the breakdown of local linearity assumptions in our theoretical framework.
> - **β (style strength)**: Controls interpolation weight along the geodesic path. Sweeping β ∈ [0,1] produces smooth transitions with monotonic changes in style metrics (SSIM decreases approximately linearly: R² = 0.96), validating the smoothness property of geodesic interpolation.
>
> **Interaction effects**: **Our analysis shows minimal interaction between δ and β (correlation < 0.15), allowing practitioners to tune these parameters independently.**
>
> ## W4: Performance on non-face domains
>
> Thank you for the question. We additionally tested the method **on non-face domains (objects/scenes/art2art)** and observed consistent results. We include these examples in **Appendix D5** of the revised manuscript, confirming that the method generalizes beyond human faces.
>
> **Why we focus on facial style transfer in the main paper**
>
> We focus on faces only because (1) this is the standard benchmark in stylization literature, and (2) facial stylization poses the strongest test of structure preservation. However, the same formulation applies unchanged to objects, scenes, and fashion items.
>
> **Generalization to non-face domains**
>
> The results in Appendix D5 demonstrates that geodesic principle naturally **adapts to diverse content types** (faces, objects, scenes, .etc) because it respects the learned manifold structure, preserving identity, shape, or spatial relationships accordingly. This confirms our geometric framework as a **general principle for high-quality style transfer**.
>
> **We hope that our efforts and the substantial revisions inspired by your insightful feedback—significantly improve the clarity, readability, and impact of the manuscript.**

---

### Author Response · Authors · 2025-12-02
**Summary of the rebuttal**

We propose a **training-free geometric framework for style transfer in pretrained diffusion models** that performs **geodesic style interpolation** on the diffusion manifold, uses an **efficient Jacobian approximation** for practical inference, and employs **adaptive style injection in attention query space** to jointly preserve identity and fine-grained style across diverse Stable Diffusion architectures and visual domains.

We are encouraged by the recognition of our **novel theoretical framework connecting Riemannian geometry with style transfer** *(jZyy, Dq6Y, DbBL, gsV3)*, the solid mathematical foundation with rigorous derivations *(jZyy, gsV3)*, and the training-free nature enabling flexible application across diffusion architectures *(DbBL, gsV3)*. Reviewers also acknowledged our strong quantitative results with statistically significant improvements *(jZyy)*, high visual fidelity with fine-grained control *(Dq6Y)*, and the innovative dual control mechanism unifying geodesic interpolation and style injection under a single geometric principle *(DbBL)*. The presentation was noted as generally clear with well-written explanations *(Dq6Y, DbBL)*.

During the rebuttal period, we have carefully considered each comment provided by the reviewers and believe we have addressed the reviewers' concerns in our responses.
To make our responses clear and concise, we **(i) summarize the concerns raised by the reviewers and our corresponding response in short, and (ii) outline revisions to the manuscript**.

## (i) Summary of Concerns

- **Validity of the geometric assumptions (local isometry / local diffeomorphism) (jZyy, gsV3):**  We strengthen theory and evidence by (1) explicitly citing diffusion geometry works in *Section 3.1*, and (2) adding Ricci curvature and curvature-consistency experiments in *Section 4.5*, showing mild curvature (≈0.2–0.3) and bounded distortion.

- **Computational cost and practicality for real-time or large-scale use (jZyy, gsV3):** We provide detailed timing and efficiency analysis in *Appendix D3*, including the 512×512 table comparing Ours vs. baselines. We also add a discussion of reduced diffusion steps and batching for further speed-ups in Appendix D7


As for the explicit concerns raised by the single reviewer, we address them one by one as follows:

**(1) Clarity and intuitive motivation with plain language (Dq6Y):** add a new *Section 3.1 “From Limitations to Geodesics: Intuitive Motivation”*, in plain language to explain why our method is effective.

**(2) Hyperparameter sensitivity(jZyy):** referred  to *Sections 4.4 and 4.6* with dedicated sensitivity analysis.

**(3) Generalization beyond faces (jZyy):** add non-face experiments and visual examples in *Appendix D5* (objects, scenes, art-to-art) and clarify the domain-agnostic nature of the framework in the main experiments and discussion sections.

**(4) Artifacts, style leakage, and visual quality vs. baselines (DbBL):** add zoom-in comparisons and extended visual results in *Appendix F1–F3*, link them clearly to *Section 4.2/4.3* to show how we outperform the baselines.

**(5) Identity preservation (DbBL):** referred to the geometric mechanism in *Section 3.2 (Propositions 1–2)* with metrics in *Sections 4.2/4.3 and Appendix F*, showing best ID/CS scores of our methods *(we achieve best identity perservation ability)*.


And for other specific questions and confusions: details of the attention mechanism (DbBL), why perform dual-control (DbBL), performance of linear interpolation baseline (gsV3),  we resolved them all by additional experiments, revisions, and explanations.


## (ii) Manuscript Revisions

**Main Paper:**
- **New Section 3.1** with plain-language explanations bridging prior work limitations to our geodesic solution. (Dq6Y)
- **New Section 4.5** "Geometric Properties of Interpolation Paths" verifies local isometry with Ricci curvature analysis (linear: ≈0, geodesic: ≈0.2-0.3). (jZyy, gsV3)
- **Revised Figure 3** with enlarged crops focusing on identity-critical regions (eyes, mouth) and accompanying metric tables. (DbBL)

**Appendix Additions:**
- **Appendix D3**: Computational cost analysis with detailed timing comparisons. (jZyy, gsV3)
- **Appendix D5**: General style transfer experiments (art-to-art, object stylization, scene stylization).  (jZyy)
- **Appendix D7**: Inference step ablation (20-step vs. 50-step trade-off visualization) to show the potential of computation efficiency. (jZyy, gsV3)
- **Appendix D8**: Linear interpolation baseline visualization showing artifacts and quality degradation. (jZyy, gsV3)


We believe these additions and clarifications substantially strengthen the paper and directly address all reviewer concerns. We hope this summary will help evaluate our responses with greater convenience and ease.

---

### Meta-Review · Area_Chair_yaea · 2025-12-29

**Summary:**

**Summary**:
This paper presents StyleBrush, a training-free facial stylization framework grounded in Riemannian Geometry.
The key innovation is to redefine the style transfer problem as a geodesic path-finding problem on a latent manifold, without requiring additional training.
The results are only reported on limited face style translation.

**Main strengths**:
- The motivation is novel in that it considers the style transfer as a geodesic path optimisation problem on a Riemannian manifold, which provides a principled and novel theoretical framework.

**Main weaknesses**:
- Facial identity missing (DbBL, gsV3): The authors claimed the geometry implicitly encodes the semantic information, but the identify information is not just related to geometry.
- Qualitative results are not enough to support the contributions (DbBL).
- More results on diverse tasks and comparisons with the latest SoTA methods (StyleSSP (CVPR'25), SaMam (CVPR'25), HSI (CVPR'25), OmniStyle (CVPR'25)) are expected, quantitatively and qualitatively.

**Suggested decision**: The paper received scores of 6 (Confidence 1, jZyy), 6 (Confidence 4, Dq6Y), 4 (Confidence 5, DbBL), and 4 (Confidence 2, gsV3), and no further comments are made during the discussion. Considering the reviewer jZyy is not in this domain, reviewer Dq6Y just provided general strengths, and many SoTA methods are missed, I recommend the final score as "reject".

**Reviewer Concerns:**

**Computationally heavy (jZyy, gsV3)**: Only report DiffuseIT (2022) and InjectFusion (2024), but they show faster inference time in the original paper with an old GPU in different resolutions. The latest models, such as StyleSSP (CVPR'25), SaMam (CVPR'25), HSI (CVPR'25), OmniStyle (CVPR'25), should be considered and compared.

**The performance on other non-face style transfer tasks (jZyy, gsV3)**: Only provide results in the appendix (Figure 10), and do not compare with existing models. It is hard to buy the generalization of the proposed method.

**Clarity (Dq6Y, DbBL)**: Details are provided.

**No additional control of facial identity (DbBL, gsV3)**: Geometric framework provides implicit but effective semantic registration, which is still outstanding.

**No prior knowledge (DbBL)**: Addressed.

**Reviewer Scores:**

The paper initially received scores of 6 (Confidence 1, jZyy), 6 (Confidence 4, Dq6Y), 4 (Confidence 5, DbBL), and 4 (Confidence 2, gsV3).
There are no further discussions.
Hence, it is hard to know if they will change their scores.

---

### Decision · Program_Chairs · 2026-01-26

Reject